# Decision trees as partitioning machines to characterize their generalization properties

**Jean-Samuel Leboeuf**
Department of Computer Science and Software Engineering
Université Laval, Québec, QC, Canada
`jean-samuel.leboeuf.1@ulaval.ca`

**Frédéric LeBlanc**
Department of Mathematics and Statistics
Université de Moncton, Moncton, NB, Canada
`efl7151@umoncton.ca`

**Mario Marchand**
Department of Computer Science and Software Engineering
Université Laval, Québec, QC, Canada
`mario.marchand@ift.ulaval.ca`

## Abstract

Decision trees are popular machine learning models that are simple to build and easy to interpret. Even though algorithms to learn decision trees date back to almost 50 years, key properties affecting their generalization error are still weakly bounded. Hence, we revisit binary decision trees on real-valued features from the perspective of partitions of the data. We introduce the notion of partitioning function, and we relate it to the growth function and to the VC dimension. Using this new concept, we are able to find the exact VC dimension of decision stumps, which is given by the largest integer $d$ such that $2\ell \geq \binom{d}{\lfloor \frac{d}{2} \rfloor}$, where $\ell$ is the number of real-valued features. We provide a recursive expression to bound the partitioning functions, resulting in a upper bound on the growth function of any decision tree structure. This allows us to show that the VC dimension of a binary tree structure with $N$ internal nodes is of order $N \log(N\ell)$. Finally, we elaborate a pruning algorithm based on these results that performs better than the CART algorithm on a number of datasets, with the advantage that no cross-validation is required.

## 1 Introduction

Decision trees are popular decision models that are versatile, intuitive, and thus useful in critical fields where the interpretability of a model is important. They are particularly useful when data is limited and not organized as in a sequence or a picture. This makes them a good alternative to deep neural networks in several cases.

Due to their expressive power, decision trees are prone to overfitting. To handle this problem, algorithms usually make use of practical techniques such as cross-validation in the learning or the pruning step. Unfortunately, cross-validation increases the running time of the learning algorithm and impairs the generalization of the tree when the number of training examples is small.

As an alternative, one can use learning algorithms based on generalization bounds. Indeed, this approach has proven its value in the work of Drouin et al. [2019], where decision trees were learned on a genomic dataset with success by optimizing a sample-compression-based bound. Such bounds guarantee that the true risk is bounded asymptotically with high probability (ignoring logarithmic terms) in $\widetilde{O}(\frac{k+d}{m-d})$, where $k$ is the number of errors made by the tree, $d$ is the size of a compressed sample and $m$ is the size of the initial dataset [Marchand and Sokolova, 2005].

Relative deviation bounds based on the VC dimension [Vapnik, 1998, Shawe-Taylor et al., 1998] are even tighter: in $\widetilde{O}(\frac{k+d}{m})$, where $d$ is the VC dimension of the tree. However, to be able to make use of such algorithms to learn or prune decision trees, we must have a reasonable estimate of the VC dimension of a decision tree class, given its structure. To the best of our knowledge, there currently exists no upper bound on the VC dimension nor the growth function of binary decision trees with real-valued features that share a common structure. The goal of this paper is to provide such bounds.

To do so, we introduce the idea of a *realizable partition* and define the notion of partitioning function, a concept closely related to the growth function and the VC dimension. We proceed to bound tightly the partitioning function of the class of decision stumps that can be constructed from a set of real-valued features, which leads us, through the use of graph theory, to find an *exact* expression of its VC dimension. To the best of our knowledge, this was previously unknown. We then extend our bound of the partitioning function to general binary decision tree structures, from which we derive the asymptotic behavior of the VC dimension of a tree with $N$ internal nodes. Finally, we show how these results can have practical implications by developing a pruning algorithm based on our bounds that outperforms CART [Breiman et al., 1984] on a number of datasets.

## 2   Related Work

For the case of binary features, Simon [1991] has shown that the VC dimension of binary decision trees of rank at most $r$ with $\ell$ features is given by $\sum_{i=0}^{r} \binom{\ell}{i}$. However, the set of decision trees with rank at most $r$ includes multiple tree structures that clearly possess different individual generalization properties. Later, Mansour [1997] claimed that the VC dimension of a binary decision tree with $N$ nodes and $\ell$ *binary* features is between $\Omega(N)$ and $O(N \log \ell)$, but did not provide the proof. Then, Maimon and Rokach [2002] provided a bound on the VC dimension of oblivious decision trees, which are trees such that all the nodes of a given layer make a split on the same feature.

In 2009, Aslan et al. proposed an exhaustive search algorithm to compute the VC dimension of decision trees with binary features. Results were obtained for all trees of height at most 4. Then, they used a regression approach to *estimate* the VC dimension of a tree as a function of the number of features, the number of nodes, and the VC dimension of the left and right subtrees.

More recently, Yıldız [2015] found the exact VC dimension of the class of decision stumps (*i.e.* trees with a single node) that can be constructed from a set of $\ell$ *binary* features, which is given by $\lfloor \log_2(\ell+1) \rfloor + 1$, and proved that this is a lower bound for the VC dimension of decision stumps with $\ell$ real-valued features. They then used these expressions as base cases to develop a recursive lower bound on the VC dimension of decision trees with more than one node. However, they did not provide an upper bound for the VC dimension of decision trees.

On a related topic, Gey [2018] found the exact VC dimension of axis-parallel cuts on $\ell$ real-valued features, which are a kind of one-sided decision stumps. They showed that the VC dimension of this class of functions is given by the largest integer $d$ such that $\ell \geq \binom{d}{\lfloor \frac{d}{2} \rfloor}$. As a corollary of their result, one has that the largest integer $d$ that satisfies $2\ell \geq \binom{d}{\lfloor \frac{d}{2} \rfloor}$ is an upper bound for the VC dimension of a decision stump, an observation they however do not make. Using a completely different approach, we here show that this upper bound is in fact exact. We discuss the difference between our results and theirs in Section 5.1.

Our work distinguishes itself from previous work by providing an upper bound for the VC dimension of any binary decision tree class on real-valued features. Our framework also extends to the multiclass setting, and we show that bound-based pruning algorithms are a viable alternative to CART.

# 3 Definitions and notation

Throughout this paper, each example $\mathbf{x} \in \mathcal{X} \stackrel{\text{def}}{=} \mathbb{R}^\ell$ is a vector of $\ell$ real-valued features[1]. We consider the multiclass setting with labels $y \in [n]$, where $[n] \stackrel{\text{def}}{=} \{1, \ldots, n\}$ for some integer $n$. Moreover, $S$ always stands for a sample of $m$ examples, and we let $x_j^i$ be the $i$-th feature of the $j$-th example of $S$.

Recall that any *tree* contains two types of nodes: *internal nodes*, which have one or many children, and *leaves* which do not have any children. For simplicity, internal nodes will be referred to as *nodes* (in contrast to leaves). In a *decision tree*, each leaf is associated with a class label and each node is associated with a decision rule, which redirects incoming examples to its children. Here, we are concerned with *binary decision trees* where each node has exactly two children and each decision rule concerns exactly one feature. A *decision stump* is a decision tree with only one node and two leaves. The output $t(\mathbf{x})$ of a tree $t$ on an example $\mathbf{x}$ is defined recursively as follows.

**Definition 1** (Output of a binary decision tree)**.** *If the tree $t$ is a leaf, the output $t(\mathbf{x})$, on example $\mathbf{x}$, is given by the class label associated with the leaf. Otherwise, if the tree $t$ is rooted at a node having a left subtree $t_l$ and a right subtree $t_r$ with a decision rule defined by feature $i \in [\ell]$, threshold $\theta \in \mathbb{R}$ and sign $s \in \{\pm 1\}$, then the output $t(\mathbf{x})$ is given by*

$$t(\mathbf{x}) \stackrel{\text{def}}{=} \begin{cases} t_l(\mathbf{x}) & \text{if } \mathrm{sign}(x^i - \theta) = s \\ t_r(\mathbf{x}) & \text{otherwise} \,, \end{cases}$$

*where $\mathrm{sign}(u) = +1$ if $u > 0$ and $\mathrm{sign}(u) = -1$ otherwise.*

From now on, we use $T$ to represent the class of binary decision trees with some fixed structure. In that case, the number of nodes and leaves and the underlying graph are fixed, but the parameters of the decision rules at the nodes and the class labels at the leaves are free parameters.

**Definition 2** (Partition)**.** *Given some finite set $A$, an $a$-partition $\bar{\alpha}(A)$ of $A$ is a set of $a \in \mathbb{N}$ disjoint and non-empty subsets $\alpha_j \subseteq A$, called* parts*, whose union is $A$.*

**Definition 3** (Growth function)**.** *We define the* growth function *$\tau_H$ of a hypothesis class $H \subseteq [n]^{\mathcal{X}}$ as the largest number of distinct functions that $H$ can realize on a sample $S$ of $m$ examples, i.e.*

$$\tau_H(m) \stackrel{\text{def}}{=} \max_{S:|S|=m} |\{h|_S : h \in H\}|\,, \tag{1}$$

*where $h|_S \stackrel{\text{def}}{=} (h(\mathbf{x}_1), h(\mathbf{x}_2), \ldots, h(\mathbf{x}_m))$, for $\mathbf{x}_j \in S$, is the restriction of $h$ to $S$.*

The growth function can sometimes be hard to evaluate exactly. Fortunately, in the binary classification setting, one can use the VC dimension to bound the growth function as it is often easier to estimate the former than the latter.

**Definition 4** (VC dimension)**.** *Let $H$ be a class of binary classifiers. A sample $S = \{\mathbf{x}_1, \ldots, \mathbf{x}_m\}$ is shattered by $H$ iff all possible Boolean functions on $S$ can be realized by functions $h \in H$. The VC dimension of $H$, $\mathrm{VCdim}\,H$, is defined as the maximal cardinality of a set $S$ shattered by $H$. In particular, the VC dimension of $H$ is the largest integer $d$ such that $\tau_H(d) = 2^d$.*

# 4 Partitions as a framework

Binary decision trees are traditionally defined as in Section 3. However, it is useful to represent decision trees as some kind of "partitioning machines". Indeed, consider a set $S$ of examples that is sieved through some tree, so that all examples are distributed among the leaves. Then, setting aside the labels, the set of non-empty leaves exactly satisfies the definition of a partition of $S$. However, when the leaves are labelled, if some leaves have the same label, we take the union of the identically labelled leaves to form a single part. Since we are interested in the set of distinct $a$-partitions that a tree class can realize, we need the following definition.

**Definition 5** (Realizable partition)**.** *Let $T$ be a binary decision tree class (of a fixed structure). An $a$-partition $\bar{\alpha}(S)$ of a sample $S$ is* realizable *by $T$ iff there exists some tree $t \in T$ such that*

- *For all parts $\alpha_j \in \bar{\alpha}(S)$, and for all examples $\mathbf{x}_1, \mathbf{x}_2 \in \alpha_j$, we have that $t(\mathbf{x}_1) = t(\mathbf{x}_2)$;*
- *For all distinct $\alpha_j, \alpha_k \in \bar{\alpha}(S)$, and for all $\mathbf{x}_1 \in \alpha_j, \mathbf{x}_2 \in \alpha_k$, we have that $t(\mathbf{x}_1) \neq t(\mathbf{x}_2)$.*

Hence, the set $\mathscr{P}_T^a(S)$ of all distinct $a$-partitions a tree class $T$ can realize on $S$ is obtained by considering all possible rules that we can use at each node of $T$ and all possible labelings in $[a]$ that we can assign to the leaves of $T$. We can link the growth function $\tau_T(m)$ of $T$ to $|\mathscr{P}_T^a(S)|$ as follows. Given some realizable $a$-partition $\bar{\alpha}(S)$, we have $n$ choices of label for any one part, then we have $n-1$ choices for the next one, because assigning it the same label would effectively create an $(a-1)$-partition. This process continues until no more parts or labels are left. Therefore, for any $a$-partition with $a \leq n$, one can produce $(n)_a$ distinct functions, where $(n)_a \stackrel{\text{def}}{=} n(n-1)\cdots(n-a+1)$ is the falling factorial. Consequently, the growth function $\tau_T(m)$ can be written as

$$\tau_T(m) = \max_{S : |S|=m} \sum_{a=1}^{\min\{m,n,L_T\}} (n)_a \, |\mathscr{P}_T^a(S)|, \tag{2}$$

where $L_T$ denotes the number of leaves of the tree class $T$ and where the sum goes up to $\min\{m, n, L_T\}$ so that every term in the sum stays well defined. This hints us to an important property of a tree class, that we call the *partitioning functions*.

**Definition 6** (Partitioning functions). *The $a$-partitioning function $\pi_T^a$ of a tree class $T$ is defined as the largest number of distinct $a$-partitions that $T$ can realize on a sample $S$ of $m$ examples, i.e.*

$$\pi_T^a(m) \stackrel{\text{def}}{=} \max_{S : |S|=m} |\mathscr{P}_T^a(S)|. \tag{3}$$

*Moreover, we refer to the set of all possible $a$-partitioning functions of $T$ for all integers $a \in [L_T]$, with $L_T$ being the number of leaves of $T$, as the* partitioning functions *of the tree class $T$.*

Since the maximum of a sum is less than or equal to the sum of the maxima of its summands, we have that

$$\tau_T(m) \leq \sum_{a=1}^{L_T} (n)_a \pi_T^a(m). \tag{4}$$

Moreover, we have equality whenever $n = 2$ or $L_T = 2$ since the first term of the sum of Equation (2) is always $\left|\mathscr{P}_T^1(S)\right| = 1$ for any $S$ with $m > 0$.

Having linked the partitioning functions to the growth function, we can relate them to the VC dimension in the following way. On one hand we have that the total number of $a$-partitions that exist on a set of $m$ elements is given by the Stirling number of the second kind, denoted $\left\{\begin{smallmatrix} m \\ a \end{smallmatrix}\right\}$ [Graham et al., 1989]. In particular, for $m \geq 1$, we have that $\left\{\begin{smallmatrix} m \\ 1 \end{smallmatrix}\right\} = 1$ and $\left\{\begin{smallmatrix} m \\ 2 \end{smallmatrix}\right\} = 2^{m-1} - 1$. In the binary classification setting, each of these partitions yield exactly 2 distinct functions by labeling the parts with the two available classes. Thus, $T$ can realize $2^m$ binary functions iff $T$ realizes every 1- and 2-partition on $S$. On the other hand, Definition 4 implies that a tree $T$ shatters a sample $S$ iff it can realize all $2^m$ functions on $S$. Therefore, since any tree class $T$ can realize the single 1-partition, we have that $T$ shatters a sample $S$ iff it realizes every 2-partition on $S$. Hence, the VC dimension of any tree class $T$ having at least one internal node is given by

$$\text{VCdim}\, T = \max \left\{ d : \pi_T^2(d) = 2^{d-1} - 1 \right\}. \tag{5}$$

## 5  Analysis of decision trees

In this section, we analyze the partitioning behavior of decision trees. First, we present an upper bound on the 2-partitioning function of decision stumps, which allows us to recover their *exact* VC dimension. Second, we extend our result to general tree classes, which leads us to find the asymptotic behavior of the VC dimension of a binary decision tree in terms of its number of internal nodes.

### 5.1  The class of decision stumps

As the class $T$ of decision stumps has only one root node and two leaves, the only non-trivial $a$-partitioning function of $T$ is $\pi_T^2(m)$, the maximum number of 2-partitions achievable on $m$ examples. The following theorem gives a tight upper bound of this quantity.

**Theorem 7** (Upper bound on the 2-partitioning function of decision stumps). *Let $T$ be the hypothesis class of decision stumps on examples of $\ell$ real-valued features. Then*

$$\pi_T^2(m) \leq \frac{1}{2} \sum_{k=1}^{m-1} \min \left\{ 2\ell, \binom{m}{k} \right\}, \tag{6}$$

*and this is an equality for $2\ell \leq m$, for $2\ell \geq \binom{m}{\lfloor \frac{m}{2} \rfloor}$, and for $1 \leq m \leq 7$.*

*Proof.* The proof is presented in Appendix A, and relies on a permutation representation of the decision rules as well as on graph-theoretical arguments to prove the equality for $2\ell \geq \binom{m}{\lfloor \frac{m}{2} \rfloor}$. □

We conjecture that the bound is an equality for all $m$, but it is not clear how to show this.

Let us compare the theorem with the *trivial bound* that is often used for decision stumps. The trivial bound consists in exploiting the fact that for each available feature, a stump can realize at most $m - 1$ different 2-partitions, which gives $\pi_T^2(m) \leq \ell(m-1) = (1/2) \sum_{k=1}^{m-1} 2\ell$. This yields $\tau_T(m) \leq 2 + 2\ell(m - 1)$ for the growth function. Comparing the trivial bound with Theorem 7, we see that the trivial bound becomes an equality for $2\ell \leq m$ and becomes strictly larger than the bound of Theorem 7 for $2\ell > m$. Also, the trivial bound exceeds the bound of Theorem 7 by $\ell(m-1) + 1 - 2^{m-1}$ for $2\ell \geq \binom{m}{\lfloor m/2 \rfloor}$ — a gap which is at least

$$\frac{1}{2} \sum_{k=1}^{m-1} \left[ \binom{m}{\lfloor \frac{m}{2} \rfloor} - \binom{m}{k} \right].$$

Each term of the sum being positive, the trivial bound can be *much larger* than the proposed bound.

Now that we have a tight upper bound on the 2-partitioning function of decision stumps, it is straightforward to find the *exact* VC dimension of decision stumps.

**Corollary 8** (VC dimension of decision stumps). *Let $T$ be the hypothesis class of decision stumps on examples of $\ell$ real-valued features. Then, the VC dimension of $T$ is implicitly given by solving for the largest integer $d$ that satisfies $2\ell \geq \binom{d}{\lfloor \frac{d}{2} \rfloor}$.*

*Proof.* According to Equation (5), the VC dimension is given by the largest integer $m$ such that $\pi_T^2(m) = 2^{m-1} - 1$. Theorem 7 gives an upper bound on the 2-partitioning function of decision stumps. Notice that for $2\ell \geq \binom{m}{\lfloor \frac{m}{2} \rfloor}$, this theorem simplifies to $\pi_T^2(m) = 2^{m-1} - 1$, while for $2\ell < \binom{m}{\lfloor \frac{m}{2} \rfloor}$, it implies $\pi_T^2(m) < 2^{m-1} - 1$. Since $\binom{m}{\lfloor \frac{m}{2} \rfloor}$ is a strictly increasing function of $m$, the largest integer $m$ such that $\pi_T^2(m) = 2^{m-1} - 1$ is the largest $m$ that satisfies $2\ell \geq \binom{m}{\lfloor \frac{m}{2} \rfloor}$. □

**Remark** Let us mention the similarities with the result of Gey [2018], where they find the VC dimension of axis-parallel cuts. They define axis-parallel cuts as some kind of asymmetric stump, where the left leaf is always labeled 0 and the right leaf is always labeled 1. The main difference is that the VC dimension of axis-parallel cuts is given by the largest integer $d$ that satisfies $\ell \geq \binom{d}{\lfloor \frac{d}{2} \rfloor}$ (the factor 2 is absent). Their approach is a set theoretic one, and we expect it would be hard to extend it to decision stumps, particularly for the case where $m$ is odd. Moreover, the graph theoretic approach used here (see Appendix A.3) allows us to recover a tight upper bound for the growth function (and therefore applies to the multiclass setting), while theirs does not.

## 5.2 Extension to general decision tree classes

We now provide an extension of Theorem 7 that applies to any binary decision tree class, before deriving the asymptotic behavior of the VC dimension of these classes.

**Theorem 9** (Upper bound on the $c$-partitioning function of decision trees). *Let $T$ be a binary decision tree class that can construct decision rules from $\ell$ real-valued features, and let $T_l$ and $T_r$ be the*

*hypothesis classes of its left and right subtrees. Let $L_T$ denote the number of leaves of $T$. Then, for $m \leq L_T$, we have $\pi_T^c(m) = \left\{ \begin{smallmatrix} m \\ c \end{smallmatrix} \right\}$, whereas for $m > L_T$, the c-partitioning function must satisfy*

$$\pi_T^c(m) \leq \left(\frac{1}{2}\right)^{\delta_{lr}} \sum_{k=L_{T_l}}^{m-L_{T_r}} \min\left\{2\ell, \binom{m}{k}\right\} \sum_{\substack{1 \leq a,b \leq c \\ a+b \geq c}} \binom{a}{c-b}\binom{b}{c-a}(a+b-c)! \, \pi_{T_l}^a(k)\pi_{T_r}^b(m-k), \quad (7)$$

*where $\delta_{lr} = 1$ if $T_l = T_r$, and $0$ otherwise.*

The proof is provided in Appendix B. It relies on a recursive decomposition of $\mathscr{P}_T^c(S)$ exposed at the beginning of the Appendix. Note that the inequality (7) of Theorem 9 reduces to the inequality (6) of Theorem 7 when $T$ is the class of decision stumps.

Theorem 9 can be used recursively to compute an upper bound on the VC dimension of decision trees. Indeed, starting with $m = L_T + 1$, one can evaluate the bound on $\pi_T^2(m)$ incrementally until it is less than $2^{m-1} - 1$, according to Equation (5). The algorithm is presented in Appendix D.

From this Theorem, one can find the asymptotic behavior of the VC dimension of a binary decision tree class on examples with real-valued features. It is stated in the following corollary.

**Corollary 10** (Asymptotic behavior of the VC dimension)**.** *Let $T$ be a class of binary decision trees with a structure containing $N$ internal nodes on examples of $\ell$ real-valued features. Then, $\text{VCdim}\,T \in O\left(N\log(N\ell)\right)$.*

The proof is given in Appendix C and relies on inductive arguments.

# 6 Experiments

To demonstrate the utility of our framework, we apply our results to the task of pruning a greedily learned decision tree with a structural risk minimization approach. We first describe the algorithm, then we carefully explain the methodology and the choices made, and finally we discuss the results.

## 6.1 The pruning algorithm

We base our pruning algorithm on Theorem 2.3 of Shawe-Taylor et al. [1998], which states that for any distribution $D$ over a set of $m$ examples, for any countable set of hypothesis classes $H_d$ (with growth function $\tau_{H_d}$) indexed by an integer $d$, and any distributions $p_d$ on $\mathbb{N}$ and $q_k$ on $[m]$, with probability at least $1 - \delta$, the true risk $R_D(h)$ of any predictor $h \in H_d$ is at most

$$\epsilon(m,k,d,\delta) \stackrel{\text{def}}{=} \frac{1}{m}\left(2k + 4\ln\left(\frac{4\tau_{H_d}(2m)}{\delta q_k p_d}\right)\right). \quad (8)$$

Although that theorem was originally stated for binary classification and for a sequence of nested hypothesis classes $H_d$ indexed by their VC dimension, it is also valid in the multiclass setting with zero-one loss if we use the growth function directly instead of the upper bound provided by Sauer's lemma. Furthermore, it is not necessary to have nested hypothesis classes, since the main argument of the proof uses the union bound which applies for any countable set of classes.

The goal of our pruning algorithm is to try to minimize the true risk $R_D(t)$ of a given tree $t$ by minimizing the upper bound $\epsilon$. It goes as follows. Given a greedily grown decision tree $t$, fixed distributions $q_k$ and $p_d$, and a fixed confidence parameter $\delta$, we compute the bound $\epsilon$ associated to this tree. Then, for each internal node of the tree, we prune the tree by replacing the subtree rooted at this node with a leaf and we compute the bound associated with the resulting tree. Among all such pruned trees, let $t'$ be the one that has the minimum bound value. If the bound of $t'$ is less than or equal to the bound of $t$, we discard $t$ and we keep $t'$ instead. We repeat this process until pruning the tree doesn't decrease the bound. The formal version of the algorithm is presented in Algorithm 3 of Appendix E.1.

A key distinction between our proposed algorithm and CART's cost-complexity pruning algorithm is that, for each pruning step, the cost-complexity algorithm makes the choice to prune a subtree based on local information, *i.e.* it depends only on the performance of that subtree. In contrast, our algorithm takes into account global information about the *whole* tree via its growth function.

We would like to emphasize that the bound (8) could not be used to prune trees prior to our work, since no upper bound on the growth function of decision trees was known. Our paper provides such a bound via Equations (4) and (7).

## 6.2 Methodology

We benchmark our pruning algorithm on 19 datasets taken from the UCI Machine Learning Repository [Dua and Graff, 2017]. We chose datasets suited to a classification task with exclusively real-valued features and no missing entries. Furthermore, we limited ourselves to datasets with 10 or less classes, as Equation (4) becomes computationally expensive for a large number of classes.

These datasets do not come with a defined train/test split. As such, we chose to randomly split each dataset so that the models are trained on 75% of the examples and tested on the remaining 25%. To limit the effect of the randomness of the splits, we run each experiment 25 times and we report the mean test accuracy and the standard deviation.

We compare our pruning algorithm to CART's cost-complexity algorithm as proposed by Breiman et al. [1984], as it is one of the most commonly used algorithms in practice (indeed, it is the implementation of the popular `scikit-learn` Python package). Another main reason is that it is natural to compare against the cost-complexity pruning algorithm, since it approximates the complexity of a tree via the number of leaves of the tree (which is an ad hoc educated guess), while our bounds on the growth function provide a theoretically valid quantifier of the tree's complexity.

We consider 4 models: the fully grown unpruned tree as generated by CART, the pruned tree after using the cost-complexity pruning algorithm, a modification of CART's cost-complexity pruning algorithm inspired by our work, and our pruning algorithm.

The first model we consider is the greedily learned tree, grown using the Gini index until the tree has 100% classification accuracy on the training set or reaches 40 leaves. We impose this limit since the computation times for pruning trees become prohibitive for a large number of leaves. We expect that this constraint does not affect results significantly since all three pruning algorithms considered reduce the number of leaves well below 40.

The second model is the CART tree, which prunes the tree from the first model according to chapter 3 of Breiman et al. [1984]. The idea is to assume that the true risk of a tree can be approximated via its empirical risk by adding a complexity term of the form $\alpha L_T$ to it, where $\alpha$ is a constant and $L_T$ is the number of leaves of the tree $T$. We did a 10-fold cross-validation on the training set to find $\alpha$.

The third model is a modification to CART's cost-complexity algorithm, where instead of assuming that the excess risk of a tree is controlled solely by the number of leaves (as in the CART algorithm), we suppose that the dependence is of the form $\frac{d}{m} \log \frac{m}{d}$, where $d = L_T \log(L_T \ell)$, $m$ is the number of examples and $\ell$ is the number of features. The form of the dependence is inspired by the form of bound (8), replacing the growth function by the approximation of Sauer's lemma and using the dependence of Corollary 10 for the VC dimension. The rest of the algorithm is then identical to CART.

Finally, the fourth model is the one proposed in the previous section. As parameters, we fixed $\delta = 0.05$ for all experiments. The choices of distributions $p_d$ and $q_k$ are arbitrary and should reflect our prior knowledge of the problem. We would like $p_d$ to go to 0 slowly as $d$ grows in order not to penalize large trees too severely. As we are working in a multiclass setting, we cannot use the VC dimension to index the hypothesis classes. Instead, as an approximation to the complexity index of a tree, we use the number of leaves, and we give the same probability $p_d$ to every tree with the same number of leaves. We thus choose to let $p_d = \frac{6}{\pi^2 L_{T_d}^2} \frac{1}{\mathrm{WE}(L_{T_d})}$, where $\mathrm{WE}(L_T)$ denotes the $L_T$-th Wedderburn-Etherington number [Bóna, 2015], which counts the number of structurally different binary trees with $L_T$ leaves.

We observed that, in the bound (8), the penalty accorded to the complexity of the tree is disproportionately larger that the penalty accorded to the number of errors. This is because much of the looseness of the bound comes from the growth function. Indeed, it is already an upper bound for the annealed entropy, and our bound of the growth function adds even more looseness on top of that. The distribution $q_k$ offers us a chance to compensate this fact by introducing a large penalty for the number of errors $k$. We chose $q_k$ of the form $(1-r)r^k$ for some $r < 1$, such that $\sum_k q_k$ is a

geometric series summing to 1. We made a 5-fold cross-validation of $r$ on a single dataset and we stuck with this value of $r$ for all others. We tried inverse powers of 2 for $r$ and we took the geometric mean of 10 draws as the final value. The Wine dataset from the UCI Machine Learning Repository [Dua and Graff, 2017] gave a value of $r = 2^{-13.7} \approx \frac{1}{13308}$. This choice makes the value of the bound $\epsilon$ larger; however, it allows to correct the gap between the complexity dependence and the dependence of the bound on the number of errors, which gives better results in practice.

When running the experiments, we observed that Equation (7) was computationally too expensive to be used directly because of the sum over $k$. Hence, we used the following upper bound instead

$$\pi_T^c(m) \leq \left(\frac{1}{2}\right)^{\delta_{lr}} (m - L_T) \, 2\ell \sum_{\substack{1 \leq a,b \leq c \\ a+b \geq c}} \binom{a}{c-b}\binom{b}{c-a}(a+b-c)! \; \pi_{T_l}^a(m - L_{T_r})\pi_{T_r}^b(m - L_{T_l}) \, ,$$

which simply replaces the sum over $k$ by $m - L_T$ times the greatest term of the sum. This modified expression was much faster to compute and had only a small impact on the bound $\epsilon$ because of the logarithmic dependence on the growth function. It is straightforward to modify Algorithm 1 of Appendix D to compute this looser bound.

All experiments were done in pure Python. The source code used in the experiments and to produce the tables is freely available at the address `https://github.com/jsleb333/paper-decision-trees-as-partitioning-machines`.

### 6.3 Results and discussion

Table 1 presents the results of the four models we tested. The column "Original" corresponds to the unpruned tree, the "CART" column is the original tree pruned with the cost-complexity pruning algorithm, "M-CART" is the modified CART algorithm with the complexity dependencies changed to reflect our findings and the "Ours" column is the original tree pruned with Shawe-Taylor's bound. More statistics about the models and the datasets used are gathered in Appendix E.

Table 1: Mean test accuracy and standard deviation on 25 random splits of 19 datasets taken from the UCI Machine Learning Repository [Dua and Graff, 2017]. In parenthesis is the total number of examples followed by the number of classes of the dataset. The best performances up to a 0.0025 accuracy gap are highlighted in bold.

| Dataset | Model | | | |
| --- | --- | --- | --- | --- |
| | Original | CART | M-CART | Ours |
| BCWD[a] (569, 2) | $0.928 \pm 0.024$ | $0.923 \pm 0.027$ | $0.930 \pm 0.017$ | $\mathbf{0.942 \pm 0.022}$ |
| Cardiotocography 10 (2126, 10) | $\mathbf{0.566 \pm 0.023}$ | $0.562 \pm 0.023$ | $\mathbf{0.566 \pm 0.024}$ | $\mathbf{0.567 \pm 0.022}$ |
| CMSC[b] (540, 2) | $0.903 \pm 0.024$ | $\mathbf{0.920 \pm 0.021}$ | $\mathbf{0.922 \pm 0.017}$ | $\mathbf{0.921 \pm 0.014}$ |
| CBS[c] (208, 2) | $\mathbf{0.727 \pm 0.061}$ | $0.702 \pm 0.054$ | $0.695 \pm 0.084$ | $0.724 \pm 0.053$ |
| DRD[d] (1151, 2) | $0.613 \pm 0.027$ | $0.576 \pm 0.044$ | $0.602 \pm 0.040$ | $\mathbf{0.622 \pm 0.023}$ |
| Fertility (100, 2) | $0.790 \pm 0.060$ | $\mathbf{0.878 \pm 0.051}$ | $\mathbf{0.878 \pm 0.051}$ | $0.866 \pm 0.056$ |
| Habermans Survival (306, 2) | $0.660 \pm 0.062$ | $\mathbf{0.746 \pm 0.043}$ | $0.721 \pm 0.043$ | $0.719 \pm 0.043$ |
| Image Segmentation (210, 7) | $\mathbf{0.862 \pm 0.048}$ | $0.814 \pm 0.144$ | $0.844 \pm 0.050$ | $0.858 \pm 0.050$ |
| Ionosphere (351, 2) | $0.891 \pm 0.035$ | $0.772 \pm 0.108$ | $0.867 \pm 0.057$ | $\mathbf{0.892 \pm 0.032}$ |
| Iris (150, 3) | $0.933 \pm 0.030$ | $0.860 \pm 0.139$ | $0.838 \pm 0.158$ | $\mathbf{0.937 \pm 0.028}$ |
| Parkinson (195, 2) | $0.859 \pm 0.062$ | $0.848 \pm 0.064$ | $0.858 \pm 0.065$ | $\mathbf{0.863 \pm 0.065}$ |
| Planning Relax (182, 2) | $0.595 \pm 0.075$ | $0.725 \pm 0.049$ | $\mathbf{0.729 \pm 0.048}$ | $0.595 \pm 0.075$ |
| QSAR Biodegradation (1055, 2) | $0.752 \pm 0.031$ | $0.741 \pm 0.033$ | $0.757 \pm 0.026$ | $\mathbf{0.761 \pm 0.028}$ |
| Seeds (210, 3) | $0.918 \pm 0.034$ | $0.914 \pm 0.040$ | $0.905 \pm 0.081$ | $\mathbf{0.925 \pm 0.033}$ |
| Spambase (4601, 2) | $0.844 \pm 0.027$ | $0.839 \pm 0.028$ | $0.842 \pm 0.029$ | $\mathbf{0.846 \pm 0.026}$ |
| Vertebral Column 3C (310, 3) | $0.800 \pm 0.050$ | $0.725 \pm 0.139$ | $0.804 \pm 0.046$ | $\mathbf{0.819 \pm 0.044}$ |
| WFR24[e] (5456, 4) | $\mathbf{0.995 \pm 0.002}$ | $\mathbf{0.994 \pm 0.002}$ | $\mathbf{0.994 \pm 0.002}$ | $\mathbf{0.994 \pm 0.001}$ |
| Wine (178, 3) | $\mathbf{0.908 \pm 0.041}$ | $0.902 \pm 0.045$ | $0.903 \pm 0.043$ | $0.904 \pm 0.046$ |
| Yeast (1484, 10) | $0.429 \pm 0.019$ | $0.368 \pm 0.059$ | $0.384 \pm 0.058$ | $\mathbf{0.442 \pm 0.019}$ |

[a]Breast Cancer Wisconsin Diagnostic, [b]Climate Model Simulation Crashes, [c]Connectionist Bench Sonar, [d]Diabetic Retinopathy Debrecen, [e]Wall Following Robot 24

Our algorithm performs better than or similarly to the other algorithms on 13 out of 19 datasets, and on 16 out of 19 when excluding the original unpruned tree. Furthermore, our algorithm is able to do well on datasets of different sizes: it has the best performance on the Iris dataset with only 150 examples as well as on the Spambase dataset with 4601 examples. The mean accuracy gain of our algorithm versus the CART algorithm is of 2.02%, which suggests that it could be profitable to use our bound-based algorithm to prune trees instead of CART. Another advantage of our pruning algorithm is that it is on average 19.5 times faster than the pruning process of CART, due to the fact that our algorithm does not rely on cross-validation.

While our algorithm works well in practice, it is unfortunate that the computed bound $\epsilon$ of the pruned tree is uninformative (*i.e.* greater than 1) most of the time. On the other hand, the good performances of our algorithm shows that Shawe-Taylor's bound (8) and our bound (7) capture the behavior of decision trees well, up to a possibly large constant factor.

It is interesting to see that our pruning algorithm and the CART algorithm do not perform the same trade-off; indeed, the final tree produced by CART has three times less leaves on average than the pruned tree generated by our algorithm. This suggests that CART prunes decision trees more aggressively than necessary.

As for our modified version of CART, it generally does better than the original CART algorithm (it has a mean accuracy gain of 1.20%), but it is not as good as the algorithm based on the bound, and as such is of limited interest.

# 7   Conclusion

By considering binary decision trees as partitioning machines, and introducing the set of partitioning functions of a tree class, we have found that the VC dimension of a tree class is given by the largest integer $d$ such that $\pi_T^2(d) = 2^{d-1} - 1$. Then, we found at tight upper bound on the 2-partitioning function of the class of decision stumps on $\ell$ real-valued features. This bound allowed us to find the exact VC dimension of decision stumps, which is given by the largest $d$ such that $2\ell \geq \binom{d}{\lfloor \frac{d}{2} \rfloor}$. It was then possible to extend these results to yield a recursive upper bound of the $c$-partitioning functions of any class of binary decision tree. As a corollary, we found that the VC dimenion of a tree class with $N$ internal nodes is of order $O(N \log(N\ell))$. Based on our findings, we proposed a pruning algorithm which performed better or similarly to CART on 16 out of 19 datasets, showing that our bound-based algorithm is a viable alternative to CART.

In the future, we wish to extend our framework to decision trees on categorical features. While our partitioning framework can also be applied to categorical features, there are some obstacles to overcome at first. Most notably, as opposed to the case of real-valued features, there exist multiple ways to produce splitting rules on categorical features. For example, ID3 [Quinlan, 1986] produces a subtree for each category, LightGBM [Ke et al., 2017] bundles features together, and CART [Breiman et al., 1984] examines all possible split combinations. Other techniques involve binary encodings such as one-versus-all or one-versus-one. Every such way to proceed may result in different partitioning patterns requiring different analyses. Furthermore, one must introduce new notation to be able to handle the specific feature distribution relevant to each problem, *i.e.* there could be a certain number of features that are binary, another number that are ternary, and so on for all category sizes. We think these difficulties can be resolved and we aim to do so in a subsequent paper.

# Broader Impact

This work could be profitable to machine learning practitioners that use decision trees to produce predictive models. The methods and results presented in this work are not incompatible with methods that try to correct the bias present in some datasets and with machine learning fairness methods that should be applied when the learned model attempts to make predictions on some aspects of human behaviour.

## Acknowledgments and Disclosure of Funding

This work was supported in part by NSERC Discovery grant RGPIN-2016-05942, by NSERC ES D scholarship PGSD3–505004–2017 and by NSERC BRPC scholarships BRPC-540188-2019. We are grateful to Gaël Letarte for his comments and suggestions on preliminary versions.

## Footnotes

[1] While decision trees are often used on a mixture of real-valued and categorical features, we limit the scope of this paper to real-valued features only, mainly because categorical features require a different analysis than the one presented. We discuss the obstacles that limit the direct generalization of our framework to this type of features in more detail in the conclusion.

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
