[Supplementary Material]

## A Proof of Theorem 7

Before proceeding with the proof, we introduce a convenient way to think about a node's decision rule. Recall that a node is associated with a rule described by a feature $i \in [\ell]$, a threshold $\theta \in \mathbb{R}$, and a sign $s \in \{\pm 1\}$. The sample $S = \{\mathbf{x}_1, \ldots, \mathbf{x}_m\}$ may be represented by a collection $\Sigma$ of $\ell$ permutations of $[m]$ representing the ordering of its data points according to their values for each feature, since that relative ordering encapsulates all pertinent information on the sample, from the perspective of decision trees. To be more precise, for each $i = 1, \ldots, \ell$, let $\sigma^i$ be a permutation of $[m]$ satisfying

$$x_{\sigma_1^i}^i \leq x_{\sigma_2^i}^i \leq \cdots \leq x_{\sigma_m^i}^i .$$

In general, unless the data points all have different values for a given feature, there may be many such permutations; just pick one arbitrarily.

Any node in a decision tree splits the data points in two according to a rule of the form

$$t(\mathbf{x}) = \begin{cases} t_l(\mathbf{x}) & \text{if } \text{sign}(x^i - \theta) = s \\ t_r(\mathbf{x}) & \text{otherwise.} \end{cases}$$

This corresponds to splitting the permutation

$$\sigma^i = \begin{bmatrix} \sigma_1^i & \sigma_2^i & \cdots & \sigma_m^i \end{bmatrix}$$

in two parts, sending examples $\mathbf{x}_{\sigma_j^i}$ to one subtree for $j \leq J$, and sending the rest of the examples to the other subtree, where $J$ is determined by $\theta$ and $s$. In fact, as long as the inequalities

$$x_{\sigma_1^i}^i < x_{\sigma_2^i}^i < \cdots < x_{\sigma_m^i}^i$$

are strict (all data points have different values for each feature), then all the different ways of splitting the (now unique) permutation $\sigma^i$ induce a split on the sample $S$ according to which it was defined. This situation could be called the worst-case scenario, because it allows for more distinct 2-partitions to be realized on the sample.

We split the proof in 4 parts: 1) the bound itself, 2) the equality for $2\ell \leq m$, 3) the equality for $2\ell \geq \binom{m}{\lfloor \frac{m}{2} \rfloor}$, and 4) the equality for $1 \leq m \leq 7$.

### A.1 Proof of part 1 of Theorem 7

We want to show that

$$\pi_T^2(m) \leq \frac{1}{2} \sum_{k=1}^{m-1} \min \left\{ 2\ell, \binom{m}{k} \right\} .$$

where $T$ is the class of decision stumps on $\ell$ real-valued features.

*Proof.* First, let $\mathcal{R}(S)$ be the set of 2-partitions of $S$ realizable by a single node, and notice that bounding the cardinality of $\mathcal{R}(S)$ directly gives a bound on $\pi_T^2(m)$ if the bound does not depend directly on $S$.

Let $\mathcal{R}_k(S) \subset \mathcal{R}(S)$ be the subset of 2-partitions with a part of size $k$, and notice $\mathcal{R}_k(S) = \mathcal{R}_{m-k}(S)$. Therefore, we can decompose $\mathcal{R}(S)$ into the disjoint union

$$\mathcal{R}(S) = \bigcup_{k=1}^{\lfloor \frac{m}{2} \rfloor} \mathcal{R}_k(S). \tag{9}$$

To bound $|\mathcal{R}_k(S)|$, first consider $k < \frac{m}{2}$. Every partition in $\mathcal{R}_k(S)$ is determined by a set of $k$ data points, so that $|\mathcal{R}_k(S)| \leq \binom{m}{k}$, the number of $k$-subsets of $S$. On the other hand, given a feature $i \in [\ell]$, in the worst-case scenario, we can split the permutation $\sigma^i$ after the $k$ first points or before the $k$ last points to induce 2 distinct elements of $\mathcal{R}_k(S)$. Since there are $\ell$ features, this makes a total of at most $2\ell$ realizable 2-partitions with a part of size $k$. We conclude that, for $k < \frac{m}{2}$, we have $|\mathcal{R}_k(S)| \leq \min \left\{ 2\ell, \binom{m}{k} \right\}$.

405  Now let $k = \frac{m}{2}$. Then the same arguments apply, except that the number of 2-partitions with a part of
406  size $k$ is $\frac{1}{2}\binom{m}{k}$ because each such partition contains two subsets of the same size $k$. Moreover, for the
407  same reason, the node can produce at most only one 2-partition with a part of size $k$ for each feature.
408  Thus, $|\mathcal{R}_k(S)| \leq \min\left\{\ell, \frac{1}{2}\binom{m}{k}\right\}$.

409  Combining our results, we have

$$|\mathcal{R}_k(S)| \leq \begin{cases} \min\left\{\ell, \frac{1}{2}\binom{m}{k}\right\} & \text{if } k = \frac{m}{2} \\ \min\left\{2\ell, \binom{m}{k}\right\} & \text{otherwise.} \end{cases} \tag{10}$$

410  Using Inequality (10), the symmetry $\mathcal{R}_k(S) = \mathcal{R}_{m-k}(S)$ yields

$$|\mathcal{R}(S)| = \sum_{k=1}^{\lfloor \frac{m}{2} \rfloor} |\mathcal{R}_k(S)| \leq \sum_{k=1}^{m-1} \min\left\{2\ell, \binom{m}{k}\right\}$$

411  which concludes the proof, since $|\mathcal{R}(S)|$ depends only on $m$ and not on $S$.  □

## A.2   Proof of part 2 of Theorem 7

413  *Proof.* We want to show that the bound of Theorem 7 is an equality for $2\ell \leq m$. To this end, we
414  want to show the existence of a sample $S$ such that

$$|\mathcal{R}_k(S)| = \begin{cases} \ell & \text{if } k = \frac{m}{2} \\ 2\ell & \text{otherwise.} \end{cases}$$

415  Since $2\ell \leq m$ implies $2\ell \leq \binom{m}{k}$ for all $k$, we will have

$$|\mathcal{R}(S)| = \sum_{k=1}^{\lfloor \frac{m}{2} \rfloor} |\mathcal{R}_k(S)| = \ell(m-1) = \frac{1}{2}\sum_{k=1}^{m-1} 2\ell = \frac{1}{2}\sum_{k=1}^{m-1} \min\left\{2\ell, \binom{m}{k}\right\}$$

416  which establishes that the bound of Theorem 7 is an equality.

417  Let us construct a suitable sample $S$. Consider the permutations $\sigma^1, \ldots, \sigma^\ell$ given by the rows of the
418  following permutation representation of $S$:

$$\Sigma = \begin{bmatrix} 1 & 2 & \ldots & l & 2l+1 & 2l+2 & \ldots & m & 2l & 2l-1 & \ldots & l+1 \\ 2 & 3 & \ldots & l+1 & 2l+1 & 2l+2 & \ldots & m & 1 & 2l & \ldots & l+2 \\ 3 & 4 & \ldots & l+2 & 2l+1 & 2l+2 & \ldots & m & 2 & 1 & \ldots & l+3 \\ \vdots & \vdots & \ddots & \vdots & \vdots & \vdots & \ddots & \vdots & \vdots & \vdots & \ddots & \vdots \\ l & l+1 & \ldots & 2l-1 & 2l+1 & 2l+2 & \ldots & m & l-1 & l-2 & \ldots & 2l \end{bmatrix}.$$

419  $\Sigma$ is built up from an $\ell \times \ell$ matrix on the left, an $\ell \times (m - 2\ell)$ matrix in the middle, and an $\ell \times \ell$
420  matrix on the right. In the remainder of this paragraph, a shift is a shift in the sequence $1, 2, \ldots, 2\ell$.
421  The first row of the left matrix is $1, 2, \ldots, \ell$; subsequent rows are obtained by shifting one position to
422  the right. The middle matrix has identical rows running from $2\ell + 1$ to $m$. The first row of the right
423  matrix is $2\ell, 2\ell - 1, \ldots, \ell + 1$; subsequent rows are obtained by shifting one position to the left. For
424  example, if $\ell = 3$ and $m = 9$, we have

$$\Sigma = \begin{bmatrix} 1 & 2 & 3 & 7 & 8 & 9 & 6 & 5 & 4 \\ 2 & 3 & 4 & 7 & 8 & 9 & 1 & 6 & 5 \\ 3 & 4 & 5 & 7 & 8 & 9 & 2 & 1 & 6 \end{bmatrix}.$$

425  It is clear that, for $k = 1, \ldots, \lfloor \frac{m}{2} \rfloor$, splitting any of these permutations after the first $k$ points or
426  before the last $k$ points always induces different 2-partitions with a part of size $k$ on the sample, as
427  long as the sample is chosen so that the strict inequalities

$$x_{\sigma_1^i}^i < x_{\sigma_2^i}^i < \cdots < x_{\sigma_m^i}^i$$

428  hold; it suffices to choose $x_{\sigma_j^i}^i = j$ for $i = 1, \ldots, \ell$ and $j = 1, \ldots, m$. This gives us a total of $\ell$
429  distinct 2-partitions if $k = \frac{m}{2}$ (with even $m$), and a total of $2\ell$ distinct permutations if $k < \frac{m}{2}$, as
430  required.  □

**A.3   Proof of part 3 of Theorem 7**

432  We prove part 3 of Theorem 7 by showing that for $2\ell \geq \binom{m}{\lfloor \frac{m}{2} \rfloor}$ (so that $2\ell \geq \binom{m}{k}$ for all $k$), there

433  exists a sample $S$ such that

$$|\mathcal{R}(S)| = \sum_{k=1}^{m-1} \binom{m}{k} = \sum_{k=1}^{m-1} \min\left\{2\ell, \binom{m}{k}\right\}.$$

434  We proceed in two steps. First, we show that there exists a sample $S$ of $m$ examples on which

435  every 2-partition with a part of size $\lfloor \frac{m}{2} \rfloor$ is realized by a stump, when $2\ell \geq \binom{m}{\lfloor \frac{m}{2} \rfloor}$. Second, we

436  use induction from this base case to establish the proof for all part sizes. More precisely, we show

437  that if there exists a sample $S_k$ such that a stump can realize every 2-partition with a part of size

438  $2 \leq k \leq \frac{m}{2}$, then there also exists a sample $S_{k-1}$ of the same size such that a stump can realize every

439  2-partition with a part of size $k$ *and* every 2-partition with a part of size $k - 1$.

440  Let $\Sigma$ be the permutation representation of $S$, as explained at the beginning of Appendix A. Further-

441  more, assume we are in the worst-case scenario where

$$x^i_{\sigma^i_1} < x^i_{\sigma^i_2} < \cdots < x^i_{\sigma^i_m}$$

442  for all $i \in [\ell]$. In this case, showing that every 2-partition of $S$ is realizable by a decision stump

443  is equivalent to showing that every $k$-subset of $[m]$ is attainable by splitting a permutation of $\Sigma$ in

444  two, either by splitting after the first $k$ elements or before the last $k$ elements for every possible $k$.

445  Moreover, we only need to consider $k$-subsets for $1 \leq k \leq \frac{m}{2}$ since $\mathcal{R}_k(S) = \mathcal{R}_{m-k}(S)$.

446  **Step 1.** We want to show that there exists a sample $S_{\lfloor \frac{m}{2} \rfloor}$ of $m$ examples on which every 2-partition

447  with a part of size $\lfloor \frac{m}{2} \rfloor$ is realized by a stump when $2\ell \geq \binom{m}{\lfloor \frac{m}{2} \rfloor}$, *i.e.* when $\ell \geq \left\lceil \frac{1}{2}\binom{m}{\lfloor \frac{m}{2} \rfloor} \right\rceil$. Let

448  $\Sigma_{\lfloor \frac{m}{2} \rfloor}$ be its permutation representation. Our problem is then equivalent to finding a matrix $\Sigma_{\lfloor \frac{m}{2} \rfloor}$

449  whose rows are permutations of $[m]$ such that each $\lfloor \frac{m}{2} \rfloor$-subset of $[m]$ may be found as the first $\lfloor \frac{m}{2} \rfloor$

450  elements or the last $\lfloor \frac{m}{2} \rfloor$ elements of a row of $\Sigma_{\lfloor \frac{m}{2} \rfloor}$.

451  This is easy for even $m$. Given that $\ell \geq \frac{1}{2}\binom{m}{\frac{m}{2}}$ and that there are exactly $\frac{1}{2}\binom{m}{\frac{m}{2}}$ different 2-partitions

452  of $[m]$ with a part of size $\frac{m}{2}$, we can fit them all the first $\ell$ rows of the matrix $\Sigma_{\frac{m}{2}}$ with the first $\frac{m}{2}$

453  elements of each row being the elements of the first part of each 2-partition. Then, $\Sigma_{\frac{m}{2}}$ induces a

454  sample $S_{\frac{m}{2}}$ on which every 2-partition is realizable by a stump. If $S_{\frac{m}{2}} = \{\mathbf{x}_1, \ldots, \mathbf{x}_m\}$, choosing

455  $x^i_{\rho^i_j} = j$, where the $\rho^i_j$ are the elements of the matrix $\Sigma_{\frac{m}{2}}$, suffices.

456  Now, let's see what happens when $m$ is odd. Consider the minimal case $\ell = \left\lceil \frac{1}{2}\binom{m}{\lfloor \frac{m}{2} \rfloor} \right\rceil$. We rephrase

457  our problem as a graph problem. Let the vertices of the graph $G = (V, E)$ be the $\lfloor \frac{m}{2} \rfloor$-subsets of

458  $[m]$ and only place edges between disjoint $\lfloor \frac{m}{2} \rfloor$-subsets. Now, pairs of $\lfloor \frac{m}{2} \rfloor$-subsets with an edge

459  connecting them are exactly the pairs of $\lfloor \frac{m}{2} \rfloor$-subsets of $[m]$ whose elements can occur in the same

460  row of $\Sigma_{\lfloor \frac{m}{2} \rfloor}$ (since each row is a permutation and therefore contains each element of $[m]$ exactly

461  once). The problem of constructing a suitable matrix $\Sigma_{\lfloor \frac{m}{2} \rfloor}$ becomes equivalent to showing that there

462  exists a subset of edges $M \subseteq E$ such that no two edges $e_1, e_2 \in M$ are incident to the same vertex,

463  with cardinality $|M| = \ell$ if $\binom{m}{\lfloor \frac{m}{2} \rfloor}$ is even and $|M| = \ell - 1$ if $\binom{m}{\lfloor \frac{m}{2} \rfloor}$ is odd (since in this case, one

464  $\lfloor \frac{m}{2} \rfloor$-subset of $[m]$ will have its own row in the matrix $\Sigma_{\lfloor \frac{m}{2} \rfloor}$). Such problems are called *matching*

465  problems in the field of graph theory.

466  As it turns out, the graph $G$ is known as the *Odd Graph* $O_n$ with $n = \lfloor \frac{m}{2} \rfloor$ (since $m = 2\lfloor \frac{m}{2} \rfloor + 1$

467  when $m$ is odd). According to Mütze et al. [2018], $O_n$ has at least one Hamiltonian cycle for $n = 1$

468  and for every $n \geq 3$, a Hamiltonian cycle being a cycle which goes through every vertex exactly once.

469  In particular, it has a Hamiltonian path as long as $n \neq 2$. This implies that for $n \neq 2$, there exists

470  a matching of size $\left\lfloor \frac{1}{2}\binom{m}{\lfloor \frac{m}{2} \rfloor} \right\rfloor$. Indeed, it suffices to take one such Hamiltonian path, add the first

471  edge to $M$, skip the next one, and continue adding every other edge to $M$ as we follow along the

472  path. This ensures that every vertex is incident to exactly one of the selected edges, except when the

Figure 1: The odd graph $O_2$, also commonly known as the Petersen Graph. One matching of size 5 is shown in bold red.

number of vertices is odd, in which case one vertex is left out (thus accounting for the floor function). The case $n = 2$ (which only occurs when $m = 5$) is exceptional and $O_2$ corresponds to the Petersen Graph, which has no Hamiltonian cycle. However, from Figure 1, we can see that there still exists a matching of size $\ell = \frac{1}{2} \binom{5}{\lfloor \frac{5}{2} \rfloor} = 5$.

We can easily construct a set of $\ell$ permutations which separates all $\lfloor \frac{m}{2} \rfloor$-subsets from such a matching. Pair off the $\lfloor \frac{m}{2} \rfloor$-subsets which are joined by an edge in the chosen matching $M$. Since $m$ is odd, choosing a pair of disjoint $\lfloor \frac{m}{2} \rfloor$-subsets of $[m]$ fixes $m - 1$ elements of a permutation, leaving exactly one possible element to complete it. Hence, sandwich the missing elements between each pair of $\lfloor \frac{m}{2} \rfloor$-subsets to construct the rows of $\Sigma_{\lfloor \frac{m}{2} \rfloor}$. If $\binom{m}{\lfloor \frac{m}{2} \rfloor}$ is even, we are done. Otherwise, put the last $\lfloor \frac{m}{2} \rfloor$-subset at the beginning of the $\ell$-th row of $\Sigma_{\lfloor \frac{m}{2} \rfloor}$. Lastly, if $\ell > \lceil \frac{1}{2} \binom{m}{m} \rceil$, then build the first $\ell$ rows of $\Sigma_{\lfloor \frac{m}{2} \rfloor}$ as described above and fill in the rest with arbitrary permutations. With this configuration, just like in the even case, $\Sigma_{\lfloor \frac{m}{2} \rfloor}$ induces a sample $S_{\frac{m}{2}}$ on which every 2-partition is realizable by a stump.

**Step 2.** We want to prove that given a sample $S_{\lfloor \frac{m}{2} \rfloor}$ on which every 2-partition with a part of size $\lfloor \frac{m}{2} \rfloor$ is realizable by a stump, we can construct a sample $S_1$ on which every 2-partition is realizable by a stump. We proceed inductively, showing that given a sample $S_k$ with $1 < k \leq \lfloor \frac{m}{2} \rfloor$ on which every 2-partition with a part of size $k, k+1, \ldots, \lfloor \frac{m}{2} \rfloor$ is realizable by a stump, there exists a sample $S_{k-1}$ of the same size as $S_k$ on which every 2-partition with a part of size $k - 1, k, k+1, \ldots, \lfloor \frac{m}{2} \rfloor$ is realizable by a stump.

Let $S_k$ be a sample such that no two of its instances have the same value for any feature, and let $\Sigma_k$ be its permutation representation.

Then, Lemma 11, proved below, assures us that there exists an injective map $\phi$ from the set $\binom{[m]}{k-1}$ of all $(k-1)$-subsets of $[m]$ to the set $\binom{[m]}{k}$ of all $k$-subsets of $[m]$ such that for every $(k-1)$-subset $a$, we have $a \subset \phi(a)$.

By assumption, $\Sigma_k$ separates all $k$-subsets of $[m]$, that is all $k$-subsets of $[m]$ appear either as the first $k$ elements or the last $k$ elements of a row of $\Sigma_k$. For some $a \in \binom{[m]}{k-1}$, reorder the elements of $\phi(a)$ appearing at the beginning or the end of a row of $\Sigma_k$ so that the elements of $a$ are either at the beginning or at the end of this row (according to whether the elements of $\phi(a)$ are at the beginning or at the end of the row). Notice that after this procedure, the new matrix $\Sigma_k'$ that is obtained still separates $\phi(a)$; moreover, it also separates $a$. Since the map $\phi$ is injective, we can continue this process without ever needing to reorder the same half-row twice, applying the same steps for each $a \in \binom{[m]}{k-1}$. This yields a final matrix $\Sigma_{k-1}$ which induces the desired sample $S_{k-1}$.

505    Now, since Lemma 11 is valid for $2 \leq k \leq \lfloor \frac{m}{2} \rfloor$, and because $S_{\lfloor \frac{m}{2} \rfloor}$ is a set on which every
506    2-partition with a part of size $\lfloor \frac{m}{2} \rfloor$ can be realized by a stump, one can repeat the process above until
507    $k = 2$ so that $\mathcal{R}(S_1)$ contains every 2-partition. Thus, $S_1$ is the set needed to conclude the proof.

508    **Lemma 11.** *Let $\binom{[m]}{k} \overset{\text{def}}{=} \{a \subseteq [m] : |a| = k\}$ be the set of all k-subsets of $[m]$. Then, for $1 \leq k < \frac{m}{2}$*
509    *there exists an injective mapping $\phi : \binom{[m]}{k} \to \binom{[m]}{k+1}$ such that $a \subset \phi(a)$ for all $a \in \binom{[m]}{k}$.*

510    *Proof.* Let $k$ be such that $1 \leq k < \frac{m}{2}$. Consider the bipartite graph $G = (V, E)$ whose set of vertices
511    is $V = \binom{[m]}{k} \cup \binom{[m]}{k+1}$, with an edge connecting $a \in \binom{[m]}{k}$ and $b \in \binom{[m]}{k+1}$ if and only if $a \subset b$, and no
512    other edges. The lemma is equivalent to finding a matching of $G$ which covers $\binom{[m]}{k}$ in the sense that
513    each vertex in $\binom{[m]}{k}$ is incident to an edge of the matching. We show the existence of such a matching
514    using Hall's marriage theorem (see Hall [1935]).

515    Let $W \subseteq \binom{[m]}{k}$ and consider the set $N(W)$ containing all the vertices in $\binom{[m]}{k+1}$ which are adjacent to
516    a vertex in $W$, that is all $(k + 1)$-subsets of $[m]$ which contain a $k$-subset of $[m]$ from $W$.

517    Given $a \in W$, we can make $m - k$ different $(k + 1)$-subsets containing $a$ by adding one the $m - k$
518    elements of $[m]$ not present in $a$ to it. Since we can do this for each $a \in W$, we obtain $(m - k)|W|$
519    (not necessarily all distinct) $(k + 1)$-subsets. In fact, in the worst case, when all $\binom{k+1}{k} = k + 1$
520    different $k$-subsets of some $b \in \binom{[m]}{k+1}$ are present in $W$, $b$ will be counted $k + 1$ times. Therefore
521    $(k + 1)|N(W)| \geq (m - k)|W|$. Moreover, since $1 \leq k < \frac{m}{2}$, we have $m - k \geq k + 1$. This means

$$|N(W)| \geq \frac{m - k}{k + 1} |W| \geq |W| .$$

522    Since this inequality holds for all $W \subseteq \binom{[m]}{k}$, a straightforward application of Hall's marriage
523    theorem yields a matching of $G$ which covers $\binom{[m]}{k}$ and proves the lemma. $\qquad \square$

### A.4   Proof of part 4 of Theorem 7

525    We now prove that

$$\pi_T^2(m) = \frac{1}{2} \sum_{k=1}^{m-1} \min \left\{ 2\ell, \binom{m}{k} \right\} \tag{11}$$

526    when $1 \leq m \leq 7$. To do so, consider the permutation representation $\Sigma$ of a sample $S$ as described
527    at the beginning of the Appendix. We explicitly define $\Sigma$ which induces a sample $S$ that shows
528    Equation (11) is satisfied.

529    One must understand the following matrices as follows. If $\ell$ is less than or equal to the total number
530    of rows of the matrix, build $\Sigma$ from the first $\ell$ rows. If $\ell$ is greater than the number of rows of the
531    matrix, add arbitrary permutations to fill out the rest of the rows of $\Sigma$; these do not matter because $\Sigma$
532    already separates all subsets of $[m]$ with its first $\ell$ rows.

533    • $m = 1$:

$$[1]$$

534    • $m = 2$:

$$[1 \quad 2]$$

535    • $m = 3$:

$$\begin{bmatrix} 1 & 2 & 3 \\ 1 & 3 & 2 \end{bmatrix}$$

536    • $m = 4$:

$$\begin{bmatrix} 1 & 2 & 4 & 3 \\ 2 & 3 & 1 & 4 \\ 1 & 3 & 2 & 4 \end{bmatrix}$$

- $m = 5$:

$$\begin{bmatrix} 1 & 2 & 3 & 5 & 4 \\ 2 & 3 & 4 & 1 & 5 \\ 3 & 4 & 1 & 2 & 5 \\ 1 & 3 & 5 & 2 & 4 \\ 1 & 4 & 2 & 3 & 5 \end{bmatrix}$$

- $m = 6$:

$$\begin{bmatrix} 1 & 2 & 3 & 6 & 5 & 4 \\ 2 & 3 & 4 & 1 & 6 & 5 \\ 3 & 4 & 5 & 2 & 1 & 6 \\ 1 & 3 & 6 & 5 & 4 & 2 \\ 3 & 5 & 2 & 1 & 6 & 4 \\ 5 & 1 & 4 & 3 & 2 & 6 \\ 1 & 4 & 3 & 6 & 2 & 5 \\ 3 & 6 & 5 & 1 & 2 & 4 \\ 1 & 2 & 5 & 3 & 4 & 6 \\ 1 & 3 & 5 & 2 & 4 & 6 \end{bmatrix}$$

- $m = 7$:

$$\begin{bmatrix} 1 & 2 & 3 & 4 & 5 & 6 & 7 \\ 2 & 3 & 4 & 7 & 1 & 5 & 6 \\ 3 & 4 & 7 & 6 & 2 & 1 & 5 \\ 4 & 7 & 6 & 2 & 5 & 1 & 3 \\ 1 & 4 & 3 & 7 & 6 & 2 & 5 \\ 5 & 7 & 4 & 3 & 2 & 1 & 6 \\ 3 & 7 & 5 & 6 & 1 & 2 & 4 \\ 2 & 7 & 4 & 1 & 6 & 3 & 5 \\ 2 & 6 & 3 & 7 & 1 & 4 & 5 \\ 1 & 7 & 3 & 5 & 2 & 4 & 6 \\ 3 & 6 & 7 & 1 & 2 & 4 & 5 \\ 1 & 4 & 7 & 6 & 2 & 3 & 5 \\ 1 & 2 & 7 & 3 & 4 & 5 & 6 \\ 1 & 5 & 7 & 2 & 3 & 4 & 6 \\ 1 & 6 & 7 & 2 & 3 & 4 & 5 \\ 2 & 3 & 7 & 5 & 1 & 4 & 6 \\ 2 & 5 & 7 & 4 & 3 & 6 & 1 \\ 2 & 6 & 7 & 1 & 3 & 4 & 5 \end{bmatrix}$$

## B Proof of Theorem 9

Theorem 9 relies on a proposition we expose in the following section, and we proceed with the proof thereafter.

### B.1 Formalizing decision trees as partitioning machines

In Section 4, we introduce the notion of trees as partitioning machines. We here formalize this idea by providing a recursive construction of partitions realizable by a tree class $T$.

Given a tree class $T$ and a sample $S$, let $\bar{\gamma} \overset{\text{def}}{=} \{\gamma_1, \ldots, \gamma_c\} \in \mathscr{P}_T^c(S)$ be some $c$-partition realizable by $T$ and let $\bar{\lambda} \overset{\text{def}}{=} \{\lambda, S \backslash \lambda\} \in \mathscr{R}(S)$ be a 2-partition realized by the root node which led to $\bar{\gamma}$. According to our definition 1 of a binary tree, we have that $\lambda$ is forwarded to the left subtree class $T_l$, which produces an $a$-partition $\bar{\alpha}(\lambda)$ while $S \backslash \lambda$ is sent to the right subtree class $T_r$ which produces a $b$-partition $\bar{\beta}(S \backslash \lambda)$, as pictured in Figure 2. As explained in Section 4, $\bar{\gamma}$ arises from the union of some of the leaves, therefore it also arises from the union of some of the parts in $\bar{\alpha}$ and $\bar{\beta}$. Moreover, this implies that $a + b$ must be greater or equal to $c$.

Figure 2: The root node splits the set $S$ into two parts, $\lambda$ and $S\backslash\lambda$, which are forwarded to the left subtree class $T_l$ and the right subtree class $T_r$ respectively. The subtrees produces partitions $\bar{\alpha}$ and $\bar{\beta}$, which can be combined to yield a $c$-partition $\bar{\gamma}$.

Note that, generally, there exists multiple partitions $\bar{\alpha}$, $\bar{\beta}$, and $\bar{\lambda}$ that yield the same partition $\bar{\gamma}$. As a consequence, we can also assume without loss of generality that $a \leq c$ and $b \leq c$. Indeed, by construction, any part $\gamma_j \in \bar{\gamma}$ is the result of the union of some subset of parts $\bar{\alpha}^j \subseteq \bar{\alpha}$ and some other subset of parts $\bar{\beta}^j \subseteq \bar{\beta}$. Note that $\bar{\alpha}^j$ and $\bar{\beta}^j$ can be empty, but not both at the same time. Using this notation, we have that $\gamma_j = \bigcup_{\alpha \in \bar{\alpha}^j} \alpha \cup \bigcup_{\beta \in \bar{\beta}^j} \beta$ for every $\gamma_j$. Consider the following partition $\bar{\alpha}' \stackrel{\text{def}}{=} \{\alpha'_j : \alpha'_j \stackrel{\text{def}}{=} \bigcup_{\alpha \in \bar{\alpha}^j} \alpha, \alpha'_j \neq \varnothing\}$ and define $\bar{\beta}'$ similarly. In this formulation, $\gamma_j$ is equal to $\alpha'_j$, $\beta'_j$, or $\alpha'_j \cup \beta'_j$. Moreover, the way $\bar{\alpha}'$ and $\bar{\beta}'$ are defined implies that $\bar{\alpha}' \in \mathscr{P}_{T_l}^{a'}(\lambda)$ and $\bar{\beta}' \in \mathscr{P}_{T_r}^{b'}(S\backslash\lambda)$ for $a' \stackrel{\text{def}}{=} |\bar{\alpha}'|$ and $b' \stackrel{\text{def}}{=} |\bar{\beta}'|$. Finally, this also implies that $a', b' \leq c$, as wanted.

Having now described how a realizable partition $\bar{\gamma} \in \mathscr{P}_T^c(S)$ of a tree class $T$ is related to the realizable partitions $\bar{\alpha}$ and $\bar{\beta}$ of its left and right subtrees, it is relevant to ask instead what partitions $\bar{\gamma}$ can be made given the partitions $\bar{\alpha}$ and $\bar{\beta}$. To do so, we define the following quantity.

**Definition 12** ($c$-partitions-set of pairwise unions of two partitions)**.** *Let $\bar{\alpha}$ be an $a$-partition of some set $A$ and $\bar{\beta}$ be a $b$-partition of some other set $B$, disjoint from $A$. Define the set $\mathscr{Q}^c(\bar{\alpha}, \bar{\beta})$ of $c$-partitions that can be constructed from pairwise unions of $\bar{\alpha}$ and $\bar{\beta}$ as follows:*

$$\mathscr{Q}^c(\bar{\alpha}, \bar{\beta}) \stackrel{\text{def}}{=} \{\, \bar{\gamma} : \bar{\gamma} \text{ is a } c\text{-partition of } A \cup B \text{ s.t. } \forall \gamma \in \bar{\gamma}, \exists \alpha \in \bar{\alpha}, \beta \in \bar{\beta}$$
$$\text{s.t. } \gamma = \alpha \text{ or } \gamma = \beta \text{ or } \gamma = \alpha \cup \beta \,\}. \tag{12}$$

From this definition, it follows that $\mathscr{Q}^c(\bar{\alpha}, \bar{\beta}) = \varnothing$ if $a + b < c$, $a > c$, or $b > c$. Moreover, if $\mathscr{A}^a(A)$ is some set of $a$-partitions of $A$ and $\mathscr{B}^b(B)$ is some set of $b$-partitions of $B$, we denote by

$$Q^c(\mathscr{A}^a(A), \mathscr{B}^b(B)) \stackrel{\text{def}}{=} \bigcup_{\substack{\bar{\alpha} \in \mathscr{A}^a(A), \\ \bar{\beta} \in \mathscr{B}^b(B)}} \mathscr{Q}^c(\bar{\alpha}, \bar{\beta})$$

the union set of the $\mathscr{Q}^c$.

We are now equipped to write a recursive relation of the set of partitions a tree $T$ can realize knowing the set of partitions its subtrees can realize.

**Proposition 13** ($c$-partitions-set decomposition of decision trees)**.** *Let $\mathscr{P}_T^c(S)$ be the set of $c$-partitions that a binary decision tree class $T$ can realize on a sample $S$ of $m > L_T$ examples, and let $T_l$ and $T_r$ be the hypothesis classes of its left and right subtrees. Then, the following decomposition holds.*

$$\mathscr{P}_T^c(S) = \bigcup_{\{\lambda, S\backslash\lambda\} \in \mathscr{R}(S)} \bigcup_{1 \leq a, b \leq c} \mathscr{Q}^c\left(\mathscr{P}_{T_l}^a(\lambda), \mathscr{P}_{T_r}^b(S\backslash\lambda)\right) \cup \mathscr{Q}^c\left(\mathscr{P}_{T_l}^a(S\backslash\lambda), \mathscr{P}_{T_r}^b(\lambda)\right), \tag{13}$$

*where $\mathscr{R}(S)$ denotes the set of 2-partitions the root node can realize on $S$.*

*Proof.* Let $\bar{\gamma} \in \mathscr{P}_T^c(S)$. Then, our explanations in the paragraphs above Definition 12 imply that if $\lambda$ is forwarded to $T_l$, then $\bar{\gamma} \in \mathscr{Q}^c(\mathscr{P}_{T_l}^a(\lambda), \mathscr{P}_{T_r}^b(S\backslash\lambda))$. Alternatively, if $\lambda$ is forwarded to $T_r$, then a similar reasoning shows that $\mathscr{Q}^c(\mathscr{P}_{T_l}^a(S\backslash\lambda), \mathscr{P}_{T_r}^b(\lambda))$. On the other hand, we also have discussed that $\mathscr{Q}^c(\bar{\alpha}, \bar{\beta})$ is the quantity that contains all $c$-partitions realizable from $\bar{\alpha}$ and $\bar{\beta}$. Therefore, taking the union over all the partitions realizable by $T_l$ and $T_r$ indeed gives $\mathscr{P}_T^c(S)$. $\qquad\square$

## B.2 Proof of the theore

We are now ready to prove Theorem 9.

*Proof.* We consider first the case where the number of examples $m$ is less than or equal to the number of leaves $L_T$ of the tree $T$. We want to show that there exists a sample $S$ such that $T$ can realize every $c$-partitions of $S$.

Let $S$ be a sample such that one feature takes distinct values for each of the $m$ examples. Then, one can choose for the root of $T$ the appropriate threshold on that feature such that $m_l \le L_{T_l}$ examples will be redirected to the left and $m_r \le L_{T_r}$ examples will be redirected to the right (where we have $L_{T_l} + L_{T_r} = L_T$ and $m_l + m_r = m$). Then each of the subtrees can do the required split on the same feature, with the required constraints on the number of examples that need to be redirected on each children, until that we have eventually at most one example per leaf. In that case, by choosing any labeling in $[c]$ for the leaves, the tree class $T$ can perform any $c$-partition of the $m$ examples out of the $\left\{ {m \atop c} \right\}$ possible ones. Consequently, we have $\pi_T^c(m) = \left\{ {m \atop c} \right\}$ for any tree class with $L_T \ge m$.

We now consider the case when $m > L_T$. We want to show the following inequality:

$$\pi_T^c(m) \le \left(\frac{1}{2}\right)^{\delta_{lr}} \sum_{k=L_{T_l}}^{m-L_{T_r}} \min\left\{2\ell, \binom{m}{k}\right\} \sum_{\substack{1 \le a,b \le c \\ a+b \ge c}} \binom{a}{c-b}\binom{b}{c-a}(a+b-c)! \, \pi_{T_l}^a(k)\pi_{T_r}^b(m-k), \quad (14)$$

where $\delta_{lr} = 1$ if $T_l = T_r$, and 0 otherwise.

In the following, we assume every examples of $S$ have distinct feature values, *i.e.* it is always possible to distinguish two examples using any feature. Indeed, assuming otherwise can only reduce the number of partitions that can be made on a sample, and therefore we have, for such a sample $S$ of $m$ examples, that $|P_T^c(S)| \le \pi_T^c(m)$.

We start from Proposition 13, which states

$$\mathscr{P}_T^c(S) = \bigcup_{\{\lambda, S\setminus\lambda\} \in \mathscr{R}(S)} \bigcup_{1 \le a,b \le c} \mathscr{Q}^c\left(\mathscr{P}_{T_l}^a(\lambda), \mathscr{P}_{T_r}^b(S\setminus\lambda)\right) \cup \mathscr{Q}^c\left(\mathscr{P}_{T_l}^a(S\setminus\lambda), \mathscr{P}_{T_r}^b(\lambda)\right). \quad (15)$$

Because $\mathscr{Q}^c$ is symmetric in its arguments and because the union over $a$ and $b$ is invariant under the exchange of $a$ and $b$, we have that

$$\bigcup_{a,b} \mathscr{Q}^c\left(\mathscr{P}_{T_l}^a(S\setminus\lambda), \mathscr{P}_{T_r}^b(\lambda)\right) = \bigcup_{a,b} \mathscr{Q}^c\left(\mathscr{P}_{T_r}^a(\lambda), \mathscr{P}_{T_l}^b(S\setminus\lambda)\right), \quad (16)$$

which is equivalent to say that one can exchange the subtrees instead of sending $\lambda$ to the left and to the right. Therefore, we have

$$\mathscr{P}_T^c(S) = \mathscr{A}_{lr} \cup \mathscr{A}_{rl} \quad \text{with } \mathscr{A}_{lr} \overset{\text{def}}{=} \bigcup_{\{\lambda, S\setminus\lambda\} \in \mathscr{R}(S)} \bigcup_{1 \le a,b \le c} \mathscr{Q}^c\left(\mathscr{P}_{T_l}^a(\lambda), \mathscr{P}_{T_r}^b(S\setminus\lambda)\right), \quad (17)$$

where we mute the other dependencies of $\mathscr{A}$ to alleviate the notation.

By the union bound, we have $|\mathscr{P}_T^c(S)| \le |\mathscr{A}_{lr}| + |\mathscr{A}_{rl}|$. Let us upper bound $|\mathscr{A}_{lr}|$. Observe that a single node is very similar to a decision stump. Indeed, the root partitions decomposition of Equation (9) also applies here, so that we have

$$\mathscr{A}_{lr} = \bigcup_{k=1}^{\lfloor \frac{m}{2} \rfloor} \bigcup_{\{\lambda, S\setminus\lambda\} \in \mathscr{R}_k(S)} \bigcup_{1 \le a,b \le c} \mathscr{Q}^c\left(\mathscr{P}_{T_l}^a(\lambda), \mathscr{P}_{T_r}^b(S\setminus\lambda)\right). \quad (18)$$

Then, we show that the union over $k$ can be changed to go from $L_{T_l}$ to $\min\left\{\lfloor \frac{m}{2} \rfloor, m - L_{T_r}\right\}$ without changing $\mathscr{A}_{lr}$. To do so, we need to show that for any partition $\bar{\gamma} \in \mathscr{A}_{lr}$, there exists at least one 2-partition $\bar{\lambda} = \{\lambda, S\setminus\lambda\}$ realized by the root node with $L_{T_l} \le |\lambda| \le \min\left\{\lfloor \frac{m}{2} \rfloor, m - L_{T_r}\right\}$ that leads to $\bar{\gamma}$. Indeed, assume $|\lambda| < L_{T_l}$. Because of our assumption below Equation 14, one can always modify the threshold of the root node to send $L_{T_l}$ examples in the subtree $T_l$ and modify the subtree so that every example ends up alone in a leaf (as we have shown in the first part of the present proof).

These examples can then be united into the part they belonged in $\bar{\gamma}$ to give the same partition as before. An analogous argument also holds for $|S\backslash\lambda| \geq L_{T_r}$, which implies $|\lambda| \leq m - L_{T_r}$ (since $m > L_T$ by assumption).

Letting $M_r \stackrel{\text{def}}{=} \min\left\{\left\lfloor\frac{m}{2}\right\rfloor, m - L_{T_r}\right\}$ and taking the union bound over $k$ and over $\mathcal{R}_k(S)$, one ends up with

$$|\mathscr{A}_{lr}| \leq \sum_{k=L_{T_l}}^{M_r} |\mathcal{R}_k(S)| \max_{\{\lambda, S\backslash\lambda\}\in\mathcal{R}_k(S)} \left| \bigcup_{1\leq a,b\leq c} \mathcal{Q}^c\left(\mathscr{P}_{T_l}^a(\lambda), \mathscr{P}_{T_r}^b(S\backslash\lambda)\right) \right|. \tag{19}$$

Let us evaluate the cardinality of $\mathcal{Q}^c\left(\mathscr{P}_{T_l}^a(\lambda), \mathscr{P}_{T_r}^b(S\backslash\lambda)\right)$ when $1 \leq a, b \leq c$ and $a + b \geq c$. Using the union bound over disjoint events, we have

$$\left|\mathcal{Q}^c\left(\mathscr{P}_{T_l}^a(\lambda), \mathscr{P}_{T_r}^b(S\backslash\lambda)\right)\right| = \sum_{\bar{\alpha}\in\mathscr{P}_{T_l}^a(\lambda)} \sum_{\bar{\beta}\in\mathscr{P}_{T_r}^b(S\backslash\lambda)} \left|\mathcal{Q}^c\left(\bar{\alpha}, \bar{\beta}\right)\right|. \tag{20}$$

Pick any $\bar{\alpha} \in \mathscr{P}_{T_l}^a(\lambda)$ and $\bar{\beta} \in \mathscr{P}_{T_r}^b(S\backslash\lambda)$. According to Definition 12, we must take the unions of some parts of $\bar{\alpha}$ and $\bar{\beta}$ to end up with a $c$-partition, with the constraint that the joined parts belongs to different partitions. We start with a total of $a + b$ parts and we must take the union of some pairs to end up with only $c$ parts. Taking the union of such a pair effectively reduces the total number of parts by one, therefore we must make $a + b - c$ unions. To make these unions, choose $a + b - c$ parts from $\bar{\alpha}$ and choose $a + b - c$ parts from $\bar{\beta}$ and join them. Since there is $(a + b - c)!$ ways to join those parts, we have that $\left|\mathcal{Q}^c(\bar{\alpha}, \bar{\beta})\right| = \binom{a}{c-b}\binom{b}{c-a}(a + b - c)!$. Since the cardinality of $\mathcal{Q}^c(\bar{\alpha}, \bar{\beta})$ depends only on $a$ and $b$ and not the partitions themselves, Equation (20) becomes

$$\left|\mathcal{Q}^c\left(\mathscr{P}_{T_l}^a(\lambda), \mathscr{P}_{T_r}^b(S\backslash\lambda)\right)\right| = \binom{a}{c-b}\binom{b}{c-a}(a + b - c)!\left|\mathscr{P}_{T_l}^a(\lambda)\right|\left|\mathscr{P}_{T_r}^b(S\backslash\lambda)\right|. \tag{21}$$

Going back to Equation (19), one has

$$|\mathscr{A}_{lr}| \leq \sum_{k=L_{T_l}}^{M_r} |\mathcal{R}_k(S)| \sum_{\substack{1\leq a,b,\leq c \\ a+b\geq c}} \binom{a}{c-b}\binom{b}{c-a}(a + b - c)! \max_{\{\lambda, S\backslash\lambda\}\in\mathcal{R}_k(S)} \left|\mathscr{P}_{T_l}^a(\lambda)\right|\left|\mathscr{P}_{T_r}^b(S\backslash\lambda)\right|. \tag{22}$$

Then, using Definition 6 for $\pi_T^c(m)$ yields

$$|\mathscr{A}_{lr}| \leq \sum_{k=L_{T_l}}^{M_r} |\mathcal{R}_k(S)| \sum_{\substack{1\leq a,b,\leq c \\ a+b\geq c}} \binom{a}{c-b}\binom{b}{c-a}(a + b - c)!\, \pi_{T_l}^a(k)\pi_{T_r}^b(m - k). \tag{23}$$

This expression also applies to $\mathscr{A}_{rl}$ by exchanging indices $l$ and $r$. Apply this exchange to Equation (23). Then let $k \to m - k$, and rename $a$ to $b$ and $b$ to $a$, so that we have

$$|\mathscr{A}_{rl}| \leq \sum_{k=M_l}^{m-L_{T_r}} |\mathcal{R}_k(S)| \sum_{\substack{1\leq a,b,\leq c \\ a+b\geq c}} \binom{a}{c-b}\binom{b}{c-a}(a + b - c)!\, \pi_{T_l}^a(k)\pi_{T_r}^b(m - k), \tag{24}$$

where $M_l \stackrel{\text{def}}{=} \max\left\{\left\lceil\frac{m}{2}\right\rceil, L_{T_l}\right\}$. Notice that the coefficients inside the sum over $k$ are the same in Equations (23) and (24). For convenience, let

$$C_k \stackrel{\text{def}}{=} \sum_{\substack{1\leq a,b,\leq c \\ a+b\geq c}} \binom{a}{c-b}\binom{b}{c-a}(a + b - c)!\, \pi_{T_l}^a(k)\pi_{T_r}^b(m - k), \tag{25}$$

so that $|\mathscr{A}_{lr}|$ and $|\mathscr{A}_{rl}|$ can written in the form $\sum_k |\mathcal{R}_k(S)|\, C_k$, with the only difference being the values that $k$ takes. We can now show that the sum over $k$ in Equations (23) and (24) can be put together to yield the theorem.

There are 4 cases to consider according to the values of $M_r = \min\left\{\left\lfloor\frac{m}{2}\right\rfloor, m - L_{T_r}\right\}$ and $M_l = \max\left\{\left\lceil\frac{m}{2}\right\rceil, L_{T_l}\right\}$. First, let $M_r = \left\lfloor\frac{m}{2}\right\rfloor$ and $M_l = \left\lceil\frac{m}{2}\right\rceil$. The sum over $k$ then goes from $L_{T_l}$ to

$\lfloor\frac{m}{2}\rfloor$ for $|\mathscr{A}_{lr}|$ and from $\lceil\frac{m}{2}\rceil$ to $m - L_{T_r}$ for $|\mathscr{A}_{rl}|$. Then, if $m$ is odd, both sums can be joined directly to go from $L_{T_l}$ to $m - L_{T_r}$. If $m$ is even, one has an extra term for $k = \frac{m}{2}$. Thus

$$|\mathscr{A}_{lr}| + |\mathscr{A}_{rl}| \leq \begin{cases} \displaystyle\sum_{k=L_{T_l}}^{m-L_{T_r}} |\mathscr{R}_k(S)|\, C_k & \text{if } m \text{ is odd} \\[2em] \left|\mathscr{R}_{\frac{m}{2}}(S)\right| C_{\frac{m}{2}} + \displaystyle\sum_{k=L_{T_l}}^{m-L_{T_r}} |\mathscr{R}_k(S)|\, C_k & \text{if } m \text{ is even.} \end{cases} \tag{26}$$

Using the upper bound on $|\mathscr{R}_k(S)|$ in Equation (10), the above expression simplifies to

$$|\mathscr{A}_{lr}| + |\mathscr{A}_{rl}| \leq \sum_{k=L_{T_l}}^{m-L_{T_r}} \min\left\{2\ell, \binom{m}{k}\right\} C_k, \tag{27}$$

valid for both cases.

Second, let $M_r = \min\left\{\lfloor\frac{m}{2}\rfloor, m - L_{T_r}\right\} = \lfloor\frac{m}{2}\rfloor$ and $M_l = \max\left\{\lceil\frac{m}{2}\rceil, L_{T_l}\right\} = L_{T_l}$. This implies that $L_{T_l} \geq \lceil\frac{m}{2}\rceil$. The sum over $k$ then goes from $L_{T_l}$ to $\lfloor\frac{m}{2}\rfloor$ for $|\mathscr{A}_{lr}|$, which consists in exactly one term if $L_{T_l} = \frac{m}{2}$ and none otherwise. For $|\mathscr{A}_{rl}|$, the sum over $k$ goes from $L_{T_l}$ to $m - L_{T_r}$. Therefore, we have

$$|\mathscr{A}_{lr}| + |\mathscr{A}_{rl}| \leq \begin{cases} \left|\mathscr{R}_{\frac{m}{2}}(S)\right| C_{\frac{m}{2}} + \displaystyle\sum_{k=L_{T_l}}^{m-L_{T_r}} |\mathscr{R}_k(S)|\, C_k & \text{if } L_{T_l} = \frac{m}{2}. \\[2em] \displaystyle\sum_{k=L_{T_l}}^{m-L_{T_r}} |\mathscr{R}_k(S)|\, C_k & \text{otherwise} \end{cases} \tag{28}$$

Again, using the upper bound on $|\mathscr{R}_k(S)|$ in Equation (10), the above expression simplifies to

$$|\mathscr{A}_{lr}| + |\mathscr{A}_{rl}| \leq \sum_{k=L_{T_l}}^{m-L_{T_r}} \min\left\{2\ell, \binom{m}{k}\right\} C_k, \tag{29}$$

valid for both cases.

Third, let $M_r = \min\left\{\lfloor\frac{m}{2}\rfloor, m - L_{T_r}\right\} = m - L_{T_r}$ and $M_l = \max\left\{\lceil\frac{m}{2}\rceil, L_{T_l}\right\} = \lceil\frac{m}{2}\rceil$. This case is very similar to the second case, where $|\mathscr{A}_{rl}|$ consists in one or zero term instead of $|\mathscr{A}_{lr}|$. Thus, the same conclusion applies.

Fourth, let $M_r = \min\left\{\lfloor\frac{m}{2}\rfloor, m - L_{T_r}\right\} = m - L_{T_r}$ and $M_l = \max\left\{\lceil\frac{m}{2}\rceil, L_{T_l}\right\} = L_{T_l}$. This case violates our starting assumption that $m$ is greater than $L_T$. Hence, we can simply ignore this case.

Collecting our results, one concludes that for all $m > L_{T_l} + L_{T_r}$, we have

$$|\mathscr{P}_T^c(S)| \leq |\mathscr{A}_{lr}| + |\mathscr{A}_{rl}| \leq \sum_{k=L_{T_l}}^{m-L_{T_r}} \min\left\{2\ell, \binom{m}{k}\right\} C_k. \tag{30}$$

Observe that the right-hand-side of this inequality is independent of $S$. Therefore, by taking the maximum value over all sample $S$ of size $m$, we have a bound for $\pi_T^c(m)$.

One can improve this result when the left and the right subtrees are the same. Indeed, in this case $\mathscr{A}_{lr} = \mathscr{A}_{rl}$ so that $\mathscr{P}_T^c(S)$ is simply equal to $\mathscr{A}_{lr}$ according to Equation (17). Moreover, the condition that $m > L_{T_l} + L_{T_r}$ implies $L_{T_r} < \frac{m}{2}$, so that $M_r$ is always equal to $\lfloor\frac{m}{2}\rfloor$. Equation (23) then becomes

$$|\mathscr{A}_{lr}| \leq \sum_{k=L_{T_l}}^{\lfloor\frac{m}{2}\rfloor} |\mathscr{R}_k(S)| \sum_{\substack{1 \leq a,b, \leq c \\ a+b \geq c}} \binom{a}{c-b}\binom{b}{c-a}(a+b-c)!\, \pi_{T_l}^a(k)\pi_{T_r}^b(m-k). \tag{31}$$

Using the fact that $\mathscr{R}_k(S) = \mathscr{R}_{m-k}(S)$, that $T_l = T_r$, and that the summation over $a$ and $b$ is symmetric, along with the bound of Equation (10) on $|\mathscr{R}_k(S)|$, one can show that

$$|\mathscr{P}_T^c(S)| \le |\mathscr{A}_{lr}| \le \frac{1}{2} \sum_{k=L_{T_l}}^{m-L_{T_r}} \min\left\{2\ell, \binom{m}{k}\right\} \sum_{\substack{1 \le a,b, \le c \\ a+b \ge c}} \binom{a}{c-b}\binom{b}{c-a}(a+b-c)! \; \pi_{T_l}^a(k)\pi_{T_r}^b(m-k),$$

(32)

which is different from Equation (30) by a factor of $1/2$ only.

We finally obtain the statement of the theorem if we use the indicator function $\mathbb{1}[\cdot]$ to handle into a single expression the cases when $T_l$ and $T_r$ are the same or not. $\qquad\square$

 # C  Proof of Corollary 10

We here give the proof of Corollary 10, which states that the asymptotic behavior of the VC dimension of a class $T$ of a binary decision tree with $N$ internal nodes on examples of $\ell$ real-valued features is given by $\mathrm{VCdim}\, T \in O\left(N \log(N\ell)\right)$.

*Proof.* Letting $c = 2$ in Theorem 9, using the fact that $2^{-\delta_{lr}} \leq 1$, $\min\left\{2\ell, \binom{m}{k}\right\} \leq 2\ell$ and $\pi_T^c(k) \leq \pi_T^c(m)$ for $k \leq m$, we have

$$\pi_T^2(m) \leq 2\ell(m - L_T)\left(1 + 2\pi_{T_l}^2(m) + 2\pi_{T_r}^2(m) + 2\pi_{T_l}^2(m)\pi_{T_r}^2(m)\right).$$

We show by induction that $\pi_T^2(m) \in O((m\ell)^N)$. Assume $\pi_T^2(m) \leq (Cm\ell)^N$ for some constant $C \geq 1$, and let $N_l$ and $N_r$ be the number of nodes in the left and right subtrees respectively, so that $N_l + N_r + 1 = N$. The previous equation becomes (with $m - L_T < m$)

$$\begin{aligned}
\pi_T^c(m) &\leq 2m\ell\left(1 + 2(Cm\ell)^{N_l} + 2(Cm\ell)^{N_r} + 2(Cm\ell)^{N_l}(Cm\ell)^{N_r}\right) \\
&\leq 14m\ell(Cm\ell)^{N_l+N_r},
\end{aligned}$$

which proves our claim for $C \geq 14$. Then, Equation (5) implies

$$\mathrm{VCdim}\, T \leq \max\left\{m : (Cm\ell)^N \geq 2^{m-1} - 1\right\}.$$

One can solve for the inequality $(Cm\ell)^N \geq 2^m$ instead, since this implies $(Cm\ell)^N \geq 2^{m-1} - 1$ is true too. The Lambert $W$ function [Corless et al., 1996] can give us an exact solution, which is $m \leq -\frac{N}{\ln 2}W_{-1}\left(-\frac{\ln 2}{CN\ell}\right)$. Since $-W_{-1}(-z^{-1}) \in O\left(\log z\right)$, we have that $\mathrm{VCdim}\, T \in O\left(N \log(N\ell)\right)$.

$\square$

# D  Algorithms to upper bound the VC dimension of decision tree classes

In this Appendix, we present the algorithms for obtaining an upper bound on the VC dimension of a tree class $T$. Algorithm 1 uses Theorem 9 to upper bound the $c$-partitioning function of a tree class. Algorithm 2 uses Algorithm 1 and Equation (5) to compute an upper bound on the VC dimension of a tree class.

---

**Algorithm 1:** PartitionFuncUpperBound$(T, c, m, \ell)$

**Input:** A tree class $T$, the number $c$ of parts in the partitions, the number $m$ of elements, the number $\ell$ of features.

Let $L_T$ be the number of leaves of $T$.

**if** $c > m$ *or* $c > L_T$ **then**
    Let $N \leftarrow 0$.
**else if** $c = m$ *or* $c = 1$ *or* $m = 1$ **then**
    Let $N \leftarrow 1$.
**else if** $m \leq L_T$ **then**
    Let $N \leftarrow \left\{ {m \atop c} \right\}$.
**else**
    Let $T_l$ and $T_r$ be the left and right subtree classes of $T$.
    Let $N \leftarrow 0$.
    **for** $k = L_{T_l}$ *to* $m - L_{T_r}$ **do**

$$Let\ N \leftarrow N + \min\left\{2\ell, \binom{m}{k}\right\} \sum_{a=1}^{c} \sum_{b=\max\{1, c-a\}}^{c} \binom{a}{c-b}\binom{b}{c-a}(a+b-c)!$$
$$\times\ \text{PartitionFuncUpperBound}(T_l, a, k, \ell)$$
$$\times\ \text{PartitionFuncUpperBound}(T_r, b, m-k, \ell).$$

        **if** $T_L = T_R$ **then**
            Let $N \leftarrow \frac{N}{2}$.

**Output:** $\min\left(N, \left\{ {m \atop c} \right\}\right)$.

---

**Algorithm 2:** VCdimUpperBound$(T, \ell)$

**Input:** A tree class $T$, the number $\ell$ of features.

**if** $T$ *is a leaf* **then**
    **Output:** 1

Let $m \leftarrow L_T + 1$.

**while** PartitionFuncUpperBound$(T, 2, m, \ell) \geq 2^{m-1} - 1$ **do**
    Let $m \leftarrow m + 1$.

**Output:** $m - 1$

---

Algorithm 2 can become quite inefficient because one has to compute the values of PartitionFuncUpperBound for increasing values of $m$, which may already have been computed for smaller values of $m$. It is thus suggested to store the values of PartitionFuncUpperBound computed for each $T$ and each $m$ to be more efficient.

We applied these algorithms to the first 11 non-equivalent binary decision trees when the number of features is $\ell = 10$. The bounds are presented in Figure 3. The lower bounds were obtained by the algorithm of Figure 7 of Yıldız [2015] in conjunction with our exact value for the VC dimension of a decision stump. Using our base case improves considerably the lower bound found by Yıldız [2015].

VCdim $T = 1$   VCdim $T = 6$   $7 \leq$ VCdim $T \leq 16$   $12 \leq$ VCdim $T \leq 21$   $8 \leq$ VCdim $T \leq 25$

$13 \leq$ VCdim $T \leq 31$   $13 \leq$ VCdim $T \leq 32$   $14 \leq$ VCdim $T \leq 40$   $18 \leq$ VCdim $T \leq 38$

$19 \leq$ VCdim $T \leq 47$   $24 \leq$ VCdim $T \leq 52$

Figure 3: Lower and upper bounds on the VC dimension of the first 11 non-equivalent trees for $\ell = 10$ real-valued features. Diamond shaped nodes are leaves while circles denote internal nodes.

 # E   Supplementary materials about the experiments

 In this Appendix, we provide more details about the experiments that were done.

 ## E.1   The pruning algorithm

 The full formal pruning algorithm is given in Algorithm 3.

---

**Algorithm 3:** PruneTreeWithBound$(t, \epsilon, \delta, m)$

**Input:** A fully grown tree $t$, a bound funtion $\epsilon$ on the true risk, a confidence internal $\delta$, the number of examples $m$.

Let $T_d$ be the tree class of the tree $t$ with complexity index $d$.
Let $k_t$ be the number of errors made by $t$.
Let $b \leftarrow \epsilon(m, k_t, d, \delta)$ according to Equation (8).
Let $B \leftarrow b$ be the final bound.
**while** $t$ *is not a leaf* **do**
    **for** every internal node $n$ of the tree $t$ **do**
        Let $t_n$ be the tree $t$ with node $n$ replaced by a leaf.
        Let $T_{d_n}$ be the tree class of the tree $t_n$ with complexity index $d_n$.
        Let $k_{t_n}$ be the number of errors made by $t_n$.
        **if** $\epsilon(m, k_{t_n}, d_n, \delta) \leq b$ **then**
            Let $b \leftarrow \epsilon(m, k_{t_n}, d_n, \delta)$ be the new best bound.
            Let $t' \leftarrow t_n$ be the new best tree.
    **if** $b \leq B$ **then**
        Let $t \leftarrow t'$.
    **else**
        **break**

**Output:** The pruned tree $t$, the associated bound $B$.

---

 ## E.2   More statistics about model performances

 We here give more statics on the performances of the model tested, such as the training accuracy, the
 number of leaves and the height of the final tree, the time it took to prune the original tree, and the
 computed bound in the case of our pruning algorithm. For each table, the caption gives the dataset
 name, the total number of examples it contains, the number of features each example has as well
 as the number of classes to predict. For more details about the table columns, see the methodology
 section 6.2.

 All experiments were run on a Intel Core i5-750 CPU running Windows 10, with 12 Go of RAM.

Table 2: Breast Cancer Wisconsin Diagnostic Dataset (569 examples, 30 features, 2 classes)

|  | Original | CART | M-CART | Ours |
|---|---|---|---|---|
| Train acc. | $1.000 \pm 0.000$ | $0.962 \pm 0.024$ | $0.965 \pm 0.020$ | $0.983 \pm 0.005$ |
| Test acc. | $0.928 \pm 0.024$ | $0.923 \pm 0.027$ | $0.930 \pm 0.017$ | $\mathbf{0.942 \pm 0.022}$ |
| Leaves | $18.0 \pm 2.6$ | $5.9 \pm 3.3$ | $5.8 \pm 3.4$ | $8.3 \pm 1.4$ |
| Height | $7.0 \pm 1.0$ | $3.4 \pm 1.6$ | $3.2 \pm 1.4$ | $4.4 \pm 0.6$ |
| Time [$s$] | N/A | $5.3 \pm 0.5$ | $5.3 \pm 0.5$ | $0.1 \pm 0.0$ |
| Bound | N/A | N/A | N/A | $1.5 \pm 0.2$ |

Table 3: Cardiotocography 10 Dataset (2126 examples, 21 features, 10 classes)

|  | Original | CART | M-CART | Ours |
|---|---|---|---|---|
| Train acc. | $0.604 \pm 0.008$ | $0.582 \pm 0.014$ | $0.586 \pm 0.014$ | $0.591 \pm 0.008$ |
| Test acc. | $\mathbf{0.566 \pm 0.023}$ | $0.562 \pm 0.023$ | $\mathbf{0.566 \pm 0.024}$ | $\mathbf{0.567 \pm 0.022}$ |
| Leaves | $40.0 \pm 0.0$ | $9.0 \pm 6.2$ | $11.2 \pm 7.0$ | $11.6 \pm 2.7$ |
| Height | $15.6 \pm 2.4$ | $5.1 \pm 2.6$ | $5.9 \pm 2.7$ | $6.8 \pm 1.3$ |
| Time $[s]$ | N/A | $25.4 \pm 1.5$ | $25.6 \pm 1.6$ | $48.3 \pm 22.0$ |
| Bound | N/A | N/A | N/A | $16.8 \pm 0.3$ |

Table 4: Climate Model Simulation Crashes Dataset (540 examples, 18 features, 2 classes)

|  | Original | CART | M-CART | Ours |
|---|---|---|---|---|
| Train acc. | $1.000 \pm 0.000$ | $0.918 \pm 0.019$ | $0.941 \pm 0.022$ | $0.977 \pm 0.008$ |
| Test acc. | $0.903 \pm 0.024$ | $\mathbf{0.920 \pm 0.021}$ | $\mathbf{0.922 \pm 0.017}$ | $\mathbf{0.921 \pm 0.014}$ |
| Leaves | $21.2 \pm 3.2$ | $1.7 \pm 2.7$ | $3.8 \pm 2.9$ | $9.6 \pm 2.2$ |
| Height | $7.2 \pm 1.3$ | $0.5 \pm 1.7$ | $2.4 \pm 1.9$ | $5.2 \pm 0.8$ |
| Time $[s]$ | N/A | $4.5 \pm 0.8$ | $4.5 \pm 0.8$ | $0.2 \pm 0.1$ |
| Bound | N/A | N/A | N/A | $1.9 \pm 0.2$ |

Table 5: Connectionist Bench Sonar Dataset (208 examples, 60 features, 2 classes)

|  | Original | CART | M-CART | Ours |
|---|---|---|---|---|
| Train acc. | $1.000 \pm 0.000$ | $0.853 \pm 0.117$ | $0.877 \pm 0.120$ | $0.963 \pm 0.012$ |
| Test acc. | $\mathbf{0.727 \pm 0.061}$ | $0.702 \pm 0.054$ | $0.695 \pm 0.084$ | $0.724 \pm 0.053$ |
| Leaves | $16.4 \pm 1.7$ | $6.0 \pm 3.3$ | $7.3 \pm 4.4$ | $10.4 \pm 1.6$ |
| Height | $6.4 \pm 0.9$ | $3.3 \pm 1.7$ | $3.6 \pm 1.9$ | $5.0 \pm 0.6$ |
| Time $[s]$ | N/A | $2.8 \pm 0.3$ | $2.7 \pm 0.2$ | $0.1 \pm 0.1$ |
| Bound | N/A | N/A | N/A | $4.5 \pm 0.4$ |

Table 6: Diabetic Retinopathy Debrecen Dataset (1151 examples, 19 features, 2 classes)

|  | Original | CART | M-CART | Ours |
|---|---|---|---|---|
| Train acc. | $0.717 \pm 0.021$ | $0.598 \pm 0.062$ | $0.625 \pm 0.058$ | $0.696 \pm 0.023$ |
| Test acc. | $0.613 \pm 0.027$ | $0.576 \pm 0.044$ | $0.602 \pm 0.040$ | $\mathbf{0.622 \pm 0.023}$ |
| Leaves | $40.0 \pm 0.0$ | $2.6 \pm 1.9$ | $4.3 \pm 4.4$ | $16.5 \pm 4.3$ |
| Height | $10.7 \pm 1.1$ | $1.5 \pm 1.6$ | $2.4 \pm 2.4$ | $7.8 \pm 1.5$ |
| Time $[s]$ | N/A | $10.6 \pm 0.7$ | $10.7 \pm 0.6$ | $2.6 \pm 0.8$ |
| Bound | N/A | N/A | N/A | $13.1 \pm 0.8$ |

Table 7: Fertility Dataset (100 examples, 9 features, 2 classes)

|  | Original | CART | M-CART | Ours |
|---|---|---|---|---|
| Train acc. | $0.992 \pm 0.007$ | $0.886 \pm 0.025$ | $0.881 \pm 0.017$ | $0.888 \pm 0.027$ |
| Test acc. | $0.790 \pm 0.060$ | $\mathbf{0.878 \pm 0.051}$ | $\mathbf{0.878 \pm 0.051}$ | $0.866 \pm 0.056$ |
| Leaves | $14.6 \pm 2.2$ | $1.4 \pm 1.4$ | $1.3 \pm 0.5$ | $1.6 \pm 1.4$ |
| Height | $6.9 \pm 1.1$ | $0.4 \pm 1.4$ | $0.3 \pm 0.5$ | $0.5 \pm 1.3$ |
| Time $[s]$ | N/A | $0.7 \pm 0.1$ | $0.6 \pm 0.1$ | $0.1 \pm 0.1$ |
| Bound | N/A | N/A | N/A | $5.0 \pm 0.7$ |

Table 8: Habermans Survival Dataset (306 examples, 3 features, 2 classes)

|            | Original          | CART              | M-CART            | Ours              |
|------------|-------------------|-------------------|-------------------|-------------------|
| Train acc. | $0.832 \pm 0.024$ | $0.732 \pm 0.015$ | $0.750 \pm 0.021$ | $0.760 \pm 0.025$ |
| Test acc.  | $0.660 \pm 0.062$ | $\mathbf{0.746 \pm 0.043}$ | $0.721 \pm 0.043$ | $0.719 \pm 0.043$ |
| Leaves     | $40.0 \pm 0.0$    | $1.0 \pm 0.0$     | $3.1 \pm 1.9$     | $3.4 \pm 1.8$     |
| Height     | $12.4 \pm 1.5$    | $0.0 \pm 0.0$     | $2.0 \pm 1.7$     | $2.1 \pm 1.4$     |
| Time [$s$] | N/A               | $4.4 \pm 0.3$     | $4.4 \pm 0.3$     | $1.6 \pm 0.2$     |
| Bound      | N/A               | N/A               | N/A               | $10.1 \pm 0.8$    |

Table 9: Image Segmentation Dataset (210 examples, 19 features, 7 classes)

|            | Original          | CART              | M-CART            | Ours              |
|------------|-------------------|-------------------|-------------------|-------------------|
| Train acc. | $1.000 \pm 0.000$ | $0.936 \pm 0.126$ | $0.960 \pm 0.035$ | $0.964 \pm 0.010$ |
| Test acc.  | $\mathbf{0.862 \pm 0.048}$ | $0.814 \pm 0.144$ | $0.844 \pm 0.050$ | $0.858 \pm 0.050$ |
| Leaves     | $17.0 \pm 1.4$    | $10.8 \pm 3.3$    | $11.2 \pm 2.7$    | $10.6 \pm 1.1$    |
| Height     | $9.8 \pm 1.3$     | $7.0 \pm 1.6$     | $7.4 \pm 1.3$     | $7.4 \pm 1.0$     |
| Time [$s$] | N/A               | $2.0 \pm 0.1$     | $2.0 \pm 0.1$     | $8.0 \pm 4.6$     |
| Bound      | N/A               | N/A               | N/A               | $4.6 \pm 0.3$     |

Table 10: Ionosphere Dataset (351 examples, 34 features, 2 classes)

|            | Original          | CART              | M-CART            | Ours              |
|------------|-------------------|-------------------|-------------------|-------------------|
| Train acc. | $1.000 \pm 0.000$ | $0.809 \pm 0.132$ | $0.916 \pm 0.051$ | $0.968 \pm 0.009$ |
| Test acc.  | $\mathbf{0.891 \pm 0.035}$ | $0.772 \pm 0.108$ | $0.867 \pm 0.057$ | $\mathbf{0.892 \pm 0.032}$ |
| Leaves     | $19.6 \pm 2.0$    | $3.4 \pm 3.6$     | $5.2 \pm 3.7$     | $9.3 \pm 1.6$     |
| Height     | $9.6 \pm 2.0$     | $1.9 \pm 2.3$     | $3.2 \pm 2.1$     | $5.4 \pm 0.8$     |
| Time [$s$] | N/A               | $4.6 \pm 0.5$     | $4.6 \pm 0.5$     | $0.4 \pm 0.2$     |
| Bound      | N/A               | N/A               | N/A               | $2.8 \pm 0.2$     |

Table 11: Iris Dataset (150 examples, 4 features, 3 classes)

|            | Original          | CART              | M-CART            | Ours              |
|------------|-------------------|-------------------|-------------------|-------------------|
| Train acc. | $1.000 \pm 0.000$ | $0.923 \pm 0.116$ | $0.901 \pm 0.130$ | $0.986 \pm 0.009$ |
| Test acc.  | $0.933 \pm 0.030$ | $0.860 \pm 0.139$ | $0.838 \pm 0.158$ | $\mathbf{0.937 \pm 0.028}$ |
| Leaves     | $7.6 \pm 1.3$     | $3.9 \pm 1.5$     | $3.8 \pm 1.4$     | $4.8 \pm 1.0$     |
| Height     | $4.8 \pm 0.8$     | $2.8 \pm 1.4$     | $2.8 \pm 1.4$     | $3.6 \pm 0.8$     |
| Time [$s$] | N/A               | $0.5 \pm 0.1$     | $0.6 \pm 0.1$     | $0.0 \pm 0.0$     |
| Bound      | N/A               | N/A               | N/A               | $2.0 \pm 0.3$     |

Table 12: Parkinson Dataset (195 examples, 22 features, 2 classes)

|            | Original          | CART              | M-CART            | Ours              |
|------------|-------------------|-------------------|-------------------|-------------------|
| Train acc. | $1.000 \pm 0.000$ | $0.908 \pm 0.098$ | $0.944 \pm 0.060$ | $0.976 \pm 0.013$ |
| Test acc.  | $0.859 \pm 0.062$ | $0.848 \pm 0.064$ | $0.858 \pm 0.065$ | $\mathbf{0.863 \pm 0.065}$ |
| Leaves     | $12.7 \pm 1.8$    | $5.6 \pm 3.5$     | $6.8 \pm 3.4$     | $8.2 \pm 1.3$     |
| Height     | $5.7 \pm 1.1$     | $3.0 \pm 2.0$     | $3.6 \pm 1.6$     | $4.0 \pm 0.7$     |
| Time [$s$] | N/A               | $1.4 \pm 0.1$     | $1.4 \pm 0.1$     | $0.0 \pm 0.0$     |
| Bound      | N/A               | N/A               | N/A               | $3.1 \pm 0.4$     |

Table 13: Planning Relax Dataset (182 examples, 12 features, 2 classes)

|  | Original | CART | M-CART | Ours |
|---|---|---|---|---|
| Train acc. | $1.000 \pm 0.000$ | $0.720 \pm 0.038$ | $0.709 \pm 0.016$ | $1.000 \pm 0.000$ |
| Test acc. | $0.595 \pm 0.075$ | $0.725 \pm 0.049$ | $\mathbf{0.729 \pm 0.048}$ | $0.595 \pm 0.075$ |
| Leaves | $29.1 \pm 2.2$ | $1.7 \pm 2.6$ | $1.0 \pm 0.0$ | $29.1 \pm 2.2$ |
| Height | $11.4 \pm 2.1$ | $0.6 \pm 2.3$ | $0.0 \pm 0.0$ | $11.4 \pm 2.1$ |
| Time $[s]$ | N/A | $2.7 \pm 0.3$ | $2.0 \pm 0.2$ | $1.0 \pm 0.3$ |
| Bound | N/A | N/A | N/A | $6.5 \pm 0.1$ |

Table 14: Qsar Biodegradation Dataset (1055 examples, 41 features, 2 classes)

|  | Original | CART | M-CART | Ours |
|---|---|---|---|---|
| Train acc. | $0.834 \pm 0.022$ | $0.758 \pm 0.042$ | $0.791 \pm 0.026$ | $0.804 \pm 0.025$ |
| Test acc. | $0.752 \pm 0.031$ | $0.741 \pm 0.033$ | $0.757 \pm 0.026$ | $\mathbf{0.761 \pm 0.028}$ |
| Leaves | $40.0 \pm 0.0$ | $3.1 \pm 2.9$ | $7.0 \pm 4.8$ | $10.3 \pm 4.0$ |
| Height | $12.8 \pm 1.7$ | $1.8 \pm 1.9$ | $4.3 \pm 2.2$ | $5.7 \pm 1.8$ |
| Time $[s]$ | N/A | $16.7 \pm 1.9$ | $16.2 \pm 1.0$ | $2.9 \pm 0.8$ |
| Bound | N/A | N/A | N/A | $8.5 \pm 0.8$ |

Table 15: Seeds Dataset (210 examples, 7 features, 3 classes)

|  | Original | CART | M-CART | Ours |
|---|---|---|---|---|
| Train acc. | $1.000 \pm 0.000$ | $0.967 \pm 0.019$ | $0.964 \pm 0.057$ | $0.981 \pm 0.007$ |
| Test acc. | $0.918 \pm 0.034$ | $0.914 \pm 0.040$ | $0.905 \pm 0.081$ | $\mathbf{0.925 \pm 0.033}$ |
| Leaves | $12.0 \pm 1.8$ | $5.7 \pm 1.8$ | $6.4 \pm 2.1$ | $7.0 \pm 0.9$ |
| Height | $6.0 \pm 0.9$ | $3.9 \pm 1.1$ | $4.1 \pm 1.1$ | $4.3 \pm 0.5$ |
| Time $[s]$ | N/A | $1.1 \pm 0.1$ | $1.2 \pm 0.1$ | $0.1 \pm 0.1$ |
| Bound | N/A | N/A | N/A | $2.4 \pm 0.4$ |

Table 16: Spambase Dataset (4601 examples, 57 features, 2 classes)

|  | Original | CART | M-CART | Ours |
|---|---|---|---|---|
| Train acc. | $0.861 \pm 0.026$ | $0.846 \pm 0.026$ | $0.850 \pm 0.028$ | $0.855 \pm 0.026$ |
| Test acc. | $0.844 \pm 0.027$ | $0.839 \pm 0.028$ | $0.842 \pm 0.029$ | $\mathbf{0.846 \pm 0.026}$ |
| Leaves | $40.0 \pm 0.0$ | $7.0 \pm 5.6$ | $8.1 \pm 5.6$ | $9.6 \pm 4.0$ |
| Height | $19.8 \pm 2.4$ | $3.8 \pm 2.5$ | $4.4 \pm 2.8$ | $5.5 \pm 2.0$ |
| Time $[s]$ | N/A | $82.4 \pm 10.1$ | $81.6 \pm 10.6$ | $3.4 \pm 0.4$ |
| Bound | N/A | N/A | N/A | $6.0 \pm 1.0$ |

Table 17: Vertebral Column 3C Dataset (310 examples, 6 features, 3 classes)

|  | Original | CART | M-CART | Ours |
|---|---|---|---|---|
| Train acc. | $1.000 \pm 0.000$ | $0.784 \pm 0.181$ | $0.881 \pm 0.044$ | $0.952 \pm 0.019$ |
| Test acc. | $0.800 \pm 0.050$ | $0.725 \pm 0.139$ | $0.804 \pm 0.046$ | $\mathbf{0.819 \pm 0.044}$ |
| Leaves | $31.8 \pm 3.4$ | $6.5 \pm 7.0$ | $6.4 \pm 4.2$ | $15.6 \pm 3.2$ |
| Height | $9.8 \pm 1.3$ | $3.5 \pm 3.5$ | $4.3 \pm 2.2$ | $7.7 \pm 1.7$ |
| Time $[s]$ | N/A | $3.0 \pm 0.3$ | $2.9 \pm 0.3$ | $12.5 \pm 3.4$ |
| Bound | N/A | N/A | N/A | $4.6 \pm 0.4$ |

Table 18: Wall Following Robot 24 Dataset (5456 examples, 24 features, 4 classes)

|            | Original          | CART              | M-CART            | Ours              |
| ---------- | ----------------- | ----------------- | ----------------- | ----------------- |
| Train acc. | $1.000 \pm 0.000$ | $0.999 \pm 0.001$ | $0.999 \pm 0.001$ | $0.998 \pm 0.001$ |
| Test acc.  | $\mathbf{0.995 \pm 0.002}$ | $\mathbf{0.994 \pm 0.002}$ | $\mathbf{0.994 \pm 0.002}$ | $\mathbf{0.994 \pm 0.001}$ |
| Leaves     | $28.5 \pm 3.3$    | $22.3 \pm 4.8$    | $22.8 \pm 4.5$    | $17.8 \pm 1.5$    |
| Height     | $9.6 \pm 1.0$     | $9.3 \pm 1.3$     | $9.3 \pm 1.3$     | $8.6 \pm 1.0$     |
| Time $[s]$ | N/A               | $59.4 \pm 3.0$    | $57.6 \pm 3.1$    | $32.9 \pm 20.5$   |
| Bound      | N/A               | N/A               | N/A               | $0.3 \pm 0.0$     |

Table 19: Wine Dataset (178 examples, 13 features, 3 classes)

|            | Original          | CART              | M-CART            | Ours              |
| ---------- | ----------------- | ----------------- | ----------------- | ----------------- |
| Train acc. | $1.000 \pm 0.000$ | $0.981 \pm 0.015$ | $0.984 \pm 0.012$ | $0.989 \pm 0.010$ |
| Test acc.  | $\mathbf{0.908 \pm 0.041}$ | $0.902 \pm 0.045$ | $0.903 \pm 0.043$ | $0.904 \pm 0.046$ |
| Leaves     | $8.0 \pm 2.3$     | $5.6 \pm 1.6$     | $5.9 \pm 1.8$     | $6.3 \pm 1.2$     |
| Height     | $4.2 \pm 1.2$     | $3.2 \pm 0.6$     | $3.4 \pm 0.8$     | $3.2 \pm 0.4$     |
| Time $[s]$ | N/A               | $0.8 \pm 0.1$     | $0.8 \pm 0.1$     | $0.0 \pm 0.0$     |
| Bound      | N/A               | N/A               | N/A               | $2.2 \pm 0.6$     |

Table 20: Yeast Dataset (1484 examples, 8 features, 10 classes)

|            | Original          | CART              | M-CART            | Ours              |
| ---------- | ----------------- | ----------------- | ----------------- | ----------------- |
| Train acc. | $0.470 \pm 0.007$ | $0.370 \pm 0.057$ | $0.386 \pm 0.058$ | $0.449 \pm 0.008$ |
| Test acc.  | $0.429 \pm 0.019$ | $0.368 \pm 0.059$ | $0.384 \pm 0.058$ | $\mathbf{0.442 \pm 0.019}$ |
| Leaves     | $40.0 \pm 0.0$    | $2.0 \pm 1.1$     | $2.7 \pm 2.0$     | $6.2 \pm 1.3$     |
| Height     | $14.2 \pm 2.0$    | $1.0 \pm 1.1$     | $1.6 \pm 1.7$     | $4.1 \pm 0.9$     |
| Time $[s]$ | N/A               | $8.5 \pm 0.4$     | $8.4 \pm 0.3$     | $418.9 \pm 74.5$  |
| Bound      | N/A               | N/A               | N/A               | $22.3 \pm 0.3$    |