[Reviews · NeurIPS 2020]

Review 1

Summary and Contributions: This paper introduces partitions as a framework for studying VC dimension, obtains expressions for the partition function of decision stumps and decision trees (leading to VC bounds), and performs numerical experiments using the resulting bounds in a structural risk minimization manner. Here, partitions are of the sample points, and the partitioning function is the largest number of a-partitions (i.e., partitions of size a) that may be formed by the (tree) classifier class. The VC dimension is then related specifically to 2-partitions.

Strengths: The results seem to be interesting and the proofs non-trivial. Additionally, there is extensive empirical validation suggesting that structural risk minimization with these bounds is useful.

Weaknesses: (1) Not having read the proofs in detail, this paper seems like a good contribution. The main question I have is whether this partition framework is useful more broadly. For example, can it be used for multiclass results or with other types of classifiers?

Correctness: As far as I can tell.

Clarity: Yes.

Relation to Prior Work: Yes, prior work is discussed.

Reproducibility: Yes

Additional Feedback: (2) As someone who hasn't studied the VC dimension of trees previously, one of my initial questions would be how this would compare to VC bounds based on parametrized function classes (usually these count the number of arithmetic operations). While I don't think those lead to interesting bounds in this case, a short comment to this effect would be helpful, I think. I've read the author response and the other reviews. I will stick with my original score.


Review 2

Summary and Contributions: Authors present a different viewpoint of decision trees based on partitions of the data. It is showed that this notion of partitioning relates to the growth function and consequently to the VC dimension. Authors then provide a pruning algorithm that does not require cross-validation and attain better performance than CART.

Strengths: 1. Theoretical results are sound although I did not check all proofs thoroughly. Empirical set up is correct although somewhat limited. 2. The novelty of this work is in relating the VC dimension to the proposed partition function. It is an interesting proposition and it is theoretically appealing but the practical significance is more limited. 3. The topic and ideas are relevant to the theoretical community which makes a good paper for acceptance.

Weaknesses: * From the theoretical analysis, the main weakness might be the analysis on pure continuous features. Nowadays, it is very unlikely to have this scenario in the most challenging machine learning problems. Thus, the theoretical implications can be limited due to this. * As a non-expert in the world of decision trees. From the empirical evaluations, I am curious to know why the method was only compared to a very old algorithm such as CART. Is it the only reasonable algorithm out there for decision trees that can be comparable to the method proposed?

Correctness: Yes, theory developed seems correct, although I skipped some proofs. Empirical methodology is simple and correct.

Clarity: Paper is very well written. Easy to follow and grasp the main ideas behind.

Relation to Prior Work: There is a good comparison to existing literature, where authors stress the difference with closer work.

Reproducibility: Yes

Additional Feedback: UPDATE: I am keeping my score to 7 after reading author's response. It is certainly a good submission as is, although it would have been a stronger submission had the analysis on categorical features been included in the paper.


Review 3

Summary and Contributions: The paper analyzes the VC dimension of hypothesis classes that consist of binary decision trees with a fixed topology and varying node labels (split criteria, predictions). It provides an exact result for decision stumps and an asymptotic result for larger trees, which both depend only on the data dimension (l) and the number of split vertices in the tree.

Strengths: The paper provides exact VC dimensions for decision stumps given a data dimension and an asymptotic behavior for decision trees, given their number of internal nodes and data dimension. These are interesting results, as there seem to be none around.

Weaknesses: As already mentioned by the authors, it would be interesting to have results for categorical data. It would be interesting to get at least a small clue of what problems lie ahead, here and why the existing work does not generalize readily to this scenario.

Correctness: I could not spot any obvious mistakes in the arguments of the authors. The bounds for data dimensions 1,2,3,4 and decision stumps seem correct/plausible.

Clarity: The paper is clearly written and allows to follow the train of thought of the authors.

Relation to Prior Work: It seems worth mentioning that Ozlem Asian ; Olcay Taner Yildiz ; Ethem Alpaydin: Calculating the VC-dimension of decision trees (2009) have given an algorithm to compute the VC dimension of binary decision trees. Is there a connection between their algorithmic approach and your proof of Corr. 10? I agree with the rebuttal of the authors that the above paper is limited in scope and solves a slightly different problem. However, it still seems reasonably related.

Reproducibility: Yes

Additional Feedback: Regarding the broader impact of the paper, I feel that a more detailed sentence on the profits of practitioners using DTs might be in order. In particular, your work has implications on the power of the learner. How could a practitioner benefit from this knowledge?

[Author Response · NeurIPS 2020]

# Author response to reviews of "Decision trees as partitioning machines to characterize their generalization properties"

We first address criticisms relevant to all reviewers and then follow with individual answers.

A concern shared by all reviewers relates to the scope of the framework, particularly to the limitation that only continuous features are considered. As we mentioned in the conclusion, the case of categorical features will be the object of future work. As a matter of fact, since the submission, we have figured out how to apply the partitioning framework to decision trees with such features. The extension can handle categorical and real-valued features at the same time, so we are hoping that the proposed framework will apply to more general settings. The new results are non-trivial and require the introduction of substantially more content which would not fit in the current paper. Therefore, we plan to publish these results in a subsequent paper.

Reviewer 1:

1. About the scope of the framework: (1) The results of the paper are multiclass since we consider the growth function and not only the VC dimension. In fact, 8 out of the 19 datasets used in the experiments were multiclass tasks. (2) Unfortunately, we were not able to think of other hypothesis classes for which the framework could be applied in a useful manner. However, our intention was specifically to treat decision trees, since they are extensively used in practice.

2. About "VC bounds based on parametrized function classes": We do not think that these bounds are relevant for decision trees since they are essentially non-parametric. Indeed, decision trees are identified by the tree architecture, the decision rule used at each internal node, and the class value at each leaf. Hence, this representation does not map naturally to a set of adjustable parameters. Moreover, the VC dimension of a function class is not always related to the number of adjustable parameters. Indeed, Vapnik, in his "Statistical Learning Theory" book (section 4.11), provides examples of parametrized function classes where the VC dimension is less than, equal to, or exceeds the number of adjustable parameters.

Reviewer 2:

1. About the comparison to CART: We agree that it could have been interesting to compare against other pruning methods such as pessimistic error pruning (Breiman 1987) and error-based pruning (Quinlan 1992) or to some newer variants of them. We chose the CART algorithm (AKA cost-complexity pruning) for two main reasons: (1) Even though the CART algorithm is old, it is still widely used in practice. As a matter of fact, it is the algorithm that is implemented in the popular `scikit-learn` Python package (as explained in the online documentation of Decision Trees in section 1.10.8. Minimal Cost-Complexity Pruning). (2) Furthermore, the way the cost-complexity pruning algorithm works is by adding a penalty to the accuracy of a subtree which depends on some ad hoc notion of complexity (and finding the optimal penalty with cross-validation). Since the VC dimension and the growth function are theoretically valid quantifiers of the said complexity, it is natural to compare to this type of pruning.

Reviewer 3:

1. About the comparison to "Calculating the VC-dimension of decision trees": The paper of Aslan et al. focuses on decision trees with binary features, while our paper focuses on real-valued features, which limits the possible links between the two papers. Their exhaustive search algorithm finds the exact value of the VC dimension of trees built from binary features, but it is a brute force algorithm, exponential in the tree size. Thus, they can only apply it to trees having a maximum depth of 4. Moreover, they try to estimate the VC dimension via a regression approach, hoping it would also apply to larger trees. On the other hand, Corollary 10 gives the asymptotic behavior of the VC dimension of trees on real-valued features (which Aslan et al. do not provide).

2. About the partially addressed broader impact: We do not see in what way we have not addressed the broader impact of our work. Could you tell us what you think is missing from our discussion?

[Meta-Review · NeurIPS 2020]

The paper provides an analysis of the combinatorial properties of classes of decision trees. The reviewers found the results to be valuable, both from a theoretical perspective (novel approach in the proofs), and from a practical perspective (leading to an SRM approach to pruning).